# Silencing long-descending inter-enlargement propriospinal neurons improves hindlimb stepping after contusive spinal cord injuries

**Courtney T Shepard[1,2,3], Brandon L Brown[1,2,3], Morgan A Van Rijswijck[3,4], Rachel M Zalla[3,4], Darlene A Burke[3], Johnny R Morehouse[3], Amberly S Riegler[3], Scott R Whittemore[1,2,3,5], David SK Magnuson[1,2,3,4,5]\***

[1]Interdisciplinary Program in Translational Neuroscience, School of Interdisciplinary and Graduate Studies, University of Louisville, Louisville, United States; [2]Department of Anatomical Sciences and Neurobiology, University of Louisville, Louisville, United States; [3]Kentucky Spinal Cord Injury Research Center, University of Louisville, Louisville, United States; [4]Speed School of Engineering, University of Louisville, Louisville, United States; [5]Department of Neurological Surgery, University of Louisville, Louisville, United States

**\*For correspondence:**
dsmagn01@louisville.edu

**Competing interest:** The authors declare that no competing interests exist.

**Abstract** Spinal locomotor circuitry is comprised of rhythm generating centers, one for each limb, that are interconnected by local and long-distance propriospinal neurons thought to carry temporal information necessary for interlimb coordination and gait control. We showed previously that conditional silencing of the long ascending propriospinal neurons (LAPNs) that project from the lumbar to the cervical rhythmogenic centers (L1/L2 to C6), disrupts right-left alternation of both the forelimbs and hindlimbs without significantly disrupting other fundamental aspects of interlimb and speed-dependent coordination (Pocratsky et al., 2020). Subsequently, we showed that silencing the LAPNs after a moderate thoracic contusive spinal cord injury (SCI) resulted in better recovered locomotor function (Shepard et al., 2021). In this research advance, we focus on the descending equivalent to the LAPNs, the long descending propriospinal neurons (LDPNs) that have cell bodies at C6 and terminals at L2. We found that conditional silencing of the LDPNs in the intact adult rat resulted in a disrupted alternation of each limb pair (forelimbs and hindlimbs) and after a thoracic contusion SCI significantly improved locomotor function. These observations lead us to speculate that the LAPNs and LDPNs have similar roles in the exchange of temporal information between the cervical and lumbar rhythm generating centers, but that the partial disruption of the pathway after SCI limits the independent function of the lumbar circuitry. Silencing the LAPNs or LDPNs effectively permits or frees-up the lumbar circuitry to function independently.

## Editor's evaluation

This paper evaluates the roles of neurons arising in the spinal cord in the cervical (neck) regions that extend axons to lumbar regions that control the legs and facilitate recovery of walking ability after spinal cord injury. The paper is important because it provides evidence that neurons arising in the neck do not help the recovery of hindlimb function and in fact, mildly impair it. Most of the evidence is convincing although some limitations were noted. The data adds new information on the role of long projecting interneurons in the spinal cord affecting limb coordination during locomotion and how their silencing helps restore partial function after spinal cord injury.

## Introduction

Locomotion is a universal and robust behavior shared by almost all animals. In mammals, locomotion involves spinal circuitry that receives descending commands from supraspinal centers and peripheral input from sensory systems. The spinal locomotor circuitry is thought to consist of rhythm generating centers and pattern formation components which together comprise the central pattern generators (CPGs) for locomotion, first described by *Brown, 1911*. It is now believed that each limb has its own CPG, and that the lumbar and cervical pattern generators are interconnected by a proprio-spinal network comprised of spinal interneurons with axonal projections that interconnect the cervical and lumbar enlargements (*Reed et al., 2006*; *Brockett et al., 2013*; *Danner et al., 2017*). More specifically, LAPNs and LDPNs provide functional coupling of the two enlargements allowing precise temporal information to pass between and among the hindlimb and forelimb CPGs (*Giovanelli Barilari and Kuypers, 1969*; *Miller et al., 1975*; *English, 1979*; *Rossignol et al., 1993*; *Juvin et al., 2005*; *Juvin et al., 2012*; *Pocratsky et al., 2017*; *Shepard et al., 2021*). This intraspinal network is also thought to be important for the propagation/integration of supraspinal signals (*Courtine et al., 2008*) and sensory information (*Alstermark and Isa, 2012*) in the uninjured system.

The LDPNs, a large subset of the spinal inter-enlargement circuitry with cell bodies in the cervical enlargement and projections to the lumbar enlargement, are the current subject of anatomical and molecular investigation due to their potential involvement in coordinated patterned behaviors such as locomotion (*Flynn et al., 2017*; *Reed et al., 2006*; *Brockett et al., 2013*; *Ni et al., 2014*), as well as their potential for promoting functional recovery from SCI via formation of a de novo bridge to bypass the lesion epicenter (*Bareyre et al., 2004*; *Vavrek et al., 2006*; *Flynn et al., 2011*; *Filli et al., 2014*; *Benthall et al., 2017*).

SCI disrupts the communication between the brain and spinal cord, resulting in an immediate inability to initiate and maintain patterned weight-supported locomotion at or below the level of lesion (*Dietz and Harkema, 2004*; *Fong et al., 2009*; *Côté et al., 2017*). Even if classified as neurologically complete, most clinical SCIs are anatomically incomplete as there is some sparing of white matter at the lesion epicenter, most often the outermost rim of the lateral and ventrolateral funiculi where the LAPN and LDPN axons reside. Therefore, these neurons and their axons may comprise a percentage of the anatomically spared circuitry, thus providing a functional bridge across the injury site. Due to these anatomical characteristics, their resistance to cell death following incomplete SCI, and their presumed function in intact animals, these neurons are well-suited to participate in locomotor recovery after incomplete SCI (*Conta and Stelzner, 2004*; *Conta Steencken and Stelzner, 2010*; *Conta Steencken et al., 2011*; *Siebert et al., 2010*).

Our recently published work which targeted LAPNs before and after a low thoracic contusive SCI revealed that the LAPNs secure left-right coordination of the hindlimbs and forelimbs in a context-dependent manner during locomotion (*Pocratsky et al., 2020*; *Shepard et al., 2021*). More explicitly, when LAPNs were silenced, animals displayed right-left coordination (phase values) outside of alternation (values ~0.5), but at speeds normally associated with strict alternation and without disruption of the primary speed-dependent gait characteristics (e.g. stance/swing times). This phenotype was context-specific and only occurred when the animals were moving from point A to B, nose up on a surface with a high coefficient of friction. The phenotype disappeared (strict alternation was observed) when animals explored (nose-down), when on a low coefficient of friction surface, or when on a treadmill. We thus hypothesized that silencing LAPNs post-SCI would lead to worse interlimb coordination but found just the opposite. Silencing LAPNs restored overground stepping ability, suggesting LAPNs have a maladaptive role in recovered locomotion after SCI (*Shepard et al., 2021*). As described earlier, the anatomical location of LDPNs suggests some are spared post-SCI and could serve as a neural substrate for functional recovery (*Reed et al., 2006*; *Brockett et al., 2013*; *Pocratsky et al., 2020*). Thus, we hypothesize that LDPNs help secure forelimb-hindlimb coordination in the intact animal, and will contribute to the recovery of function post-SCI. To test this hypothesis, and to advance our work on the long propriospinal network, we conditionally silenced LDPNs in the intact animal and after recovery from an incomplete SCI.

## Results

### Silencing LDPNs alters hindlimb interlimb coordination while maintaining intralimb coordination and key locomotor features in uninjured animals

For this study, we employed a two-virus synaptic silencing strategy developed by Tadashi Isa and colleagues (*Kinoshita et al., 2012*). This system involves a highly efficient lentiviral vector (HiRet) with a tetracycline response element upstream of enhanced tetanus neurotoxin (eTeNt) and EGFP (eTeNT. EGFP) that infects neuron terminals. Next, an AAV2/2 virus that contains a tetracycline transactivator (rtTAV16) is delivered which infects neuron cell bodies. Doxycycline (Dox) then induces the production of eTeNT/EGFP which cleaves the vesicular docking protein VAMP2 thus preventing neurotransmitter release and silencing the neurons based only on their anatomy, any/only neurons with terminals exposed to the lentiviral vector and cell bodies exposed to the AAV2/2 will be silenced, as employed previously (*Pocratsky et al., 2017*; *Pocratsky et al., 2020*; *Shepard et al., 2021*). To silence the LDPNs, we performed bilateral injections at L2 and C6 spinal cord segments to doubly infect LDPNs at their terminals and cell bodies, respectively (*Figure 1A*). Only doubly-infected neurons that constitutively express rtTAV16 will then express enhanced tetanus neurotoxin (eTeNT) in the presence of Dox. At the level of the cell terminal, active eTeNT prevents synaptic vesicle release, leading to 'silenced' neurotransmission (*Figure 1B*). Removing Dox restores functional neurotransmission, allowing for reversible silencing of inter-enlargement LDPNs both before and after SCI (*Figure 1C*). We examined the coupling patterns of the limb pairs by comparing the temporal relationship, which can be expressed as a phase value. As described in our previous work (*Pocratsky et al., 2017*; *Pocratsky et al., 2020*), phase is determined by dividing the initial contact time of the trailing limb by the stride time of the leading limb. Phase values of 1 indicate synchrony and values of 0.5 indicate alternation of the limb pair. We determined the mean phase value of the limb pairs and any value >2 SD from this mean was considered 'irregular,' as defined by the blue boxes. All the locomotor assessments were done on two different walking surfaces as previously described: bare acrylic with a coefficient of friction (CoF) of 0.44 and Sylgard-coated acrylic with a CoF of 1.73, representing surfaces with relatively poor grip and extremely good grip, respectively (*Pocratsky et al., 2020*).

When the LDPNs were silenced in otherwise intact animals, we observed a disruption of right-left hindlimb and forelimb alternation that was, in many respects, similar to those observed for LAPN silencing. The two limb pairs demonstrated a substantial proportion of irregular steps and the disruption involved a similar bias towards the hindlimbs, with the forelimb pair showing less significant disruption (*Figure 2A*). However, in contrast to the highly context-dependent LAPN phenotype, LDPN silencing disrupted left-right hindlimb alternation for both the hindlimbs and forelimbs and occurred on both the low CoF (*Figure 2a–c*; small yellow circles; *Videos 1 and 2*) and the high CoF (*Figure 2A, D and E*; small red circles; *Videos 3 and 4*) surfaces. Also distinct from what we saw with the LAPNs, a second round of Dox administration led to greater disruptions in the alternation of both limb pairs, as well as the contralateral hindlimb-forelimb pair, such that there were a greater number of irregular steps during Dox2 than during Dox1, and again these disruptions occurred on both surfaces (*Figure 2B–E*; *Figure 2—figure supplement 1*; *Pocratsky et al., 2020*). During silencing, the disruption to interlimb coordination included modest but significant changes in heterolateral/contralateral and homolateral/ipsilateral forelimb-hindlimb coupling, as observed previously when silencing the LAPNs (*Figure 2—figure supplement 1*; *Pocratsky et al., 2020*). Based on the magnitude of the effect and the apparent hierarchy of impact, we consider the disruption of forelimb-hindlimb coupling to be secondary to the partial de-coupling of the forelimb and hindlimb pairs (see discussion). The silencing effects on limb pair relationships were restored when Dox was removed from the drinking water (*Figure 2B–E*, 'PostD1').

We further examined whether silencing non-SCI animals would adversely affect the coordination of the limbs in time (temporal measures) and space (spatial measures). Swing time, stance time, stride time, stride frequency, and stride distance comprise well-described gait indices that have been shown to have specific relationships with speed (*Gillis and Biewener, 2001*). As observed previously for the LAPNs, silencing LDPNs did not affect the fundamental gait relationships of swing, stance, or stride times regardless of walking surface (*Figure 3A–C and F–H*). Stride distance and frequency were also unaffected (*Figure 3D, E, I and J*). It is interesting to note that overall, even at control time points,

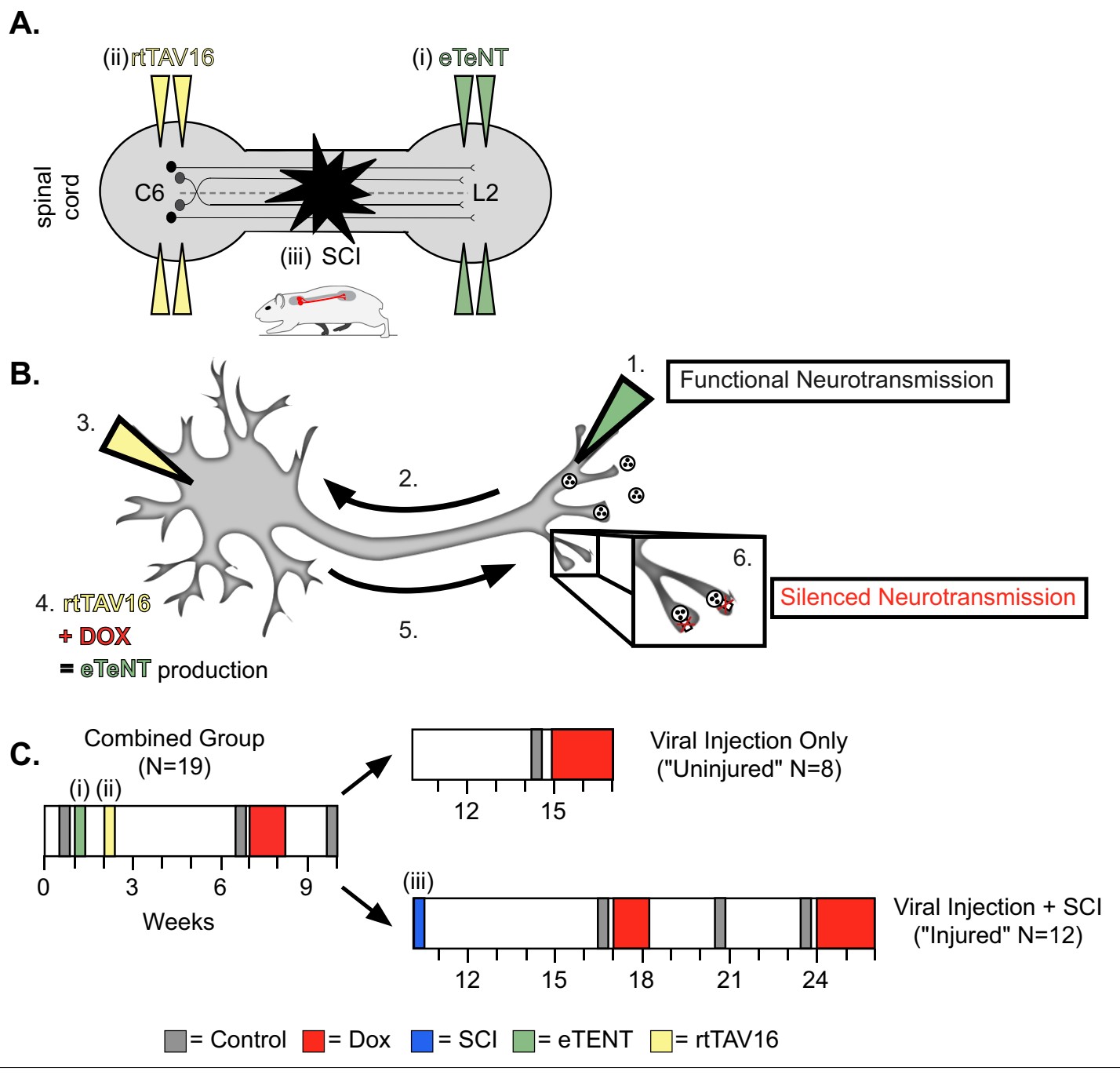

**Figure 1.** Experimental design and timeline. (**A**) Bilateral injections of enhanced tetanus neurotoxin (eTeNT) (green) and tetracycline transactivator (rtTAV16) (yellow) were performed at L2 and C6 spinal cord levels, respectively, followed by a spinal cord injury (black star). Administration of doxycycline (Dox, **B** and **C**) induces eTeNT expression in doubly-infected neurons. (**B**) When each viral vector is injected into the location of anatomically defined cell terminals (1. EGFP.eTeNT, green), and neuronal cell bodies (3. AAV2, yellow) the EGFP.eTeNT is retrogradely transported (2.) and the neuron becomes doubly-infected. rtTAV16 is activated in the presence of doxycycline (Dox, red) and can bind to the TRE promotor, which induces eTeNT. EGFP expression (4.). It is anterogradely transported down to the cell bodies (5.), where it cleaves vesicle-associated membrane protein 2 (VAMP2) and prevents the release of vesicular contents into the synapse (6. silenced neurotransmission). (**C**) Before and after viral injections (i) and (ii), pre-injury behavioral assessments were taken at three control time points (gray boxes) and a single round of Dox administration (red box). In viral injection (uninjured) controls, and following spinal cord injury (SCI) (iii), behavioral assessments were repeated at control time points (gray boxes) and during Dox administration (red boxes).

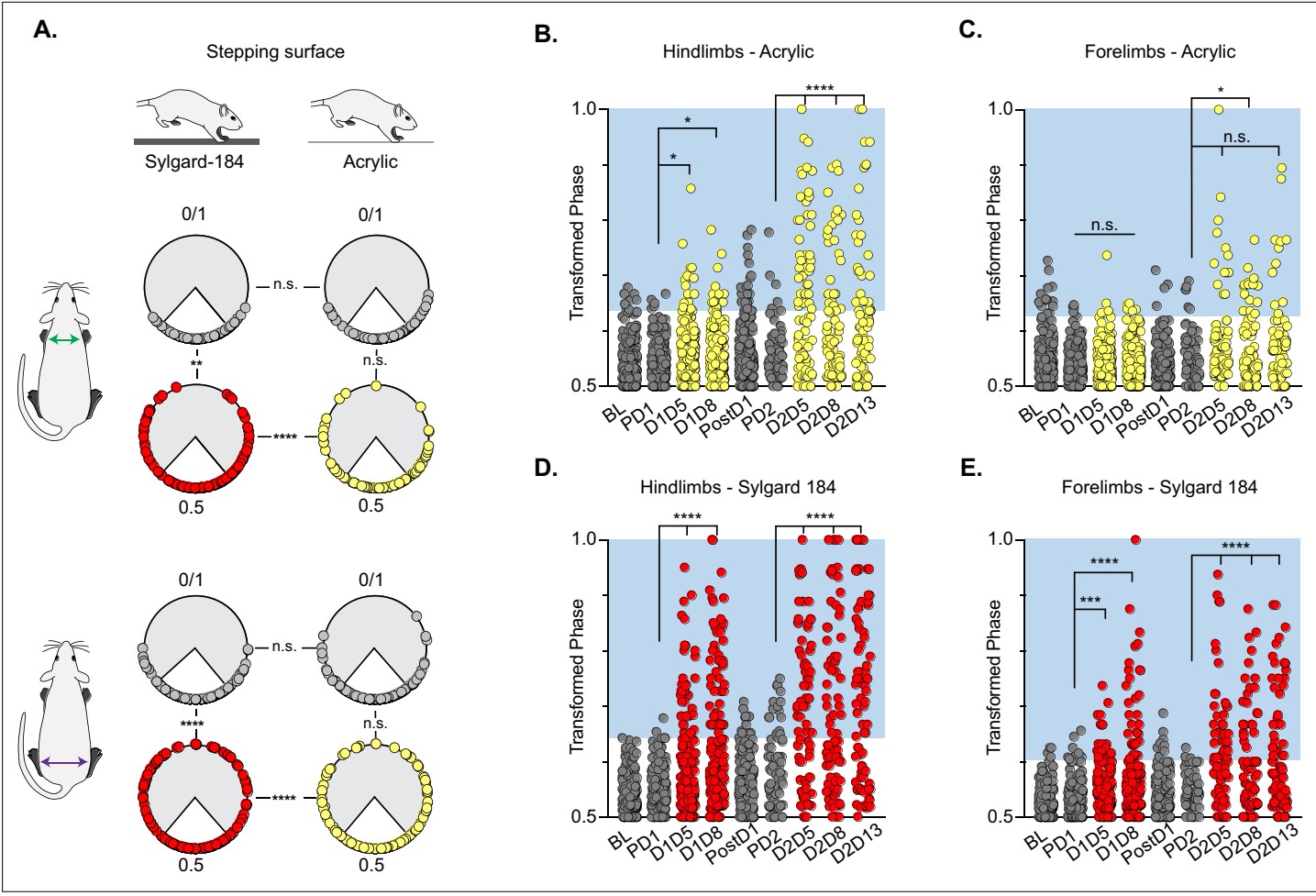

**Figure 2.** Silencing-induced perturbations for long descending propriospinal neurons (LDPNs) are not context-dependent. For Control (e.g. PD1=pre Dox 1), doxycycline (Dox) (e.g. D1D5=Dox 1 day 5), and post-Dox (PostD1) time points, all individual hindlimb and forelimb steps are graphed using circular phase plots (**A**) assessed using Watson's Non-Parametric U$^2$ circular statistics (**Table 1**). All control steps are indicated by gray circles. Acrylic (low coefficient of friction, CoF) Dox steps are indicated by yellow circles and Sylgard (high CoF) Dox steps are indicated by red circles. Hindlimb and forelimb phase relationships were transformed and graphed linearly for each time point (**B–E**). No significance was seen between the time points on the acrylic surface for the first Dox administration, but was significantly altered during the subsequent Dox administration (B, right column and B, C: PD1 hindlimbs n=3/238 [1.26%] vs D1D5 hindlimbs n=12/218 [5.50%]; *p<0.05, z=2.49; PD1 hindlimbs n=3/238 [1.26%] vs D1D8 hindlimbs n=11/218 [5.05%]; *p<0.05, z=2.29; PD2 hindlimbs n=4/82 [4.88%] vs D2D5 hindlimbs n=38/82 [46.34%]; ****p<0.001, z=6.91; PD2 hindlimbs n=4/82 [4.88%] vs D2D8 hindlimbs n=25/88 [28.41%]; ****p<0.001, z=4.39). Hindlimb stepping was significantly altered during all Dox time points on the Sylgard surface (B, left column and d, E: PD1 hindlimbs n=2/189 [1.05%] vs D1D5 hindlimbs n=37/177 [20.90%]; ****p<0.001, z=6.15; PD1 hindlimbs n=2/189 [1.05%] vs D1D8 hindlimbs n=51/171 [29.82%]; ****p<0.001, z=7.69; PD2 hindlimbs n=13/81 [16.04%] vs D2D5 hindlimbs n=44/76 [57.89%]; ****p<0.001, z=4.45; PD2 hindlimbs n=13/81 [16.04%] vs D2D8 hindlimbs n=38/80 [47.50%]; ****p<0.001, z=4.29; PD2 hindlimbs n=13/81 [16.04%] vs D2D13 hindlimbs n=48/83 [57.83%]; ****p<0.001, z=5.53). Forelimb phase relationships are shown in (**C, E**) for each time point, with less severe coordination disruptions on acrylic (C, PD1 forelimbs n=5/238 [2.10%] vs D1D5 forelimbs n=7/218 [3.21%]; n.s., z=0.73; PD1 forelimbs n=5/238 [2.10%] vs D1D8 forelimbs n=7/218 [3.21%]; n.s., z=0.73; PD2 forelimbs n=6/82 [7.32%] vs D2D5 forelimbs n=14/82 [17.07%]; n.s., z=1.93; PD2 forelimbs n=6/82 [7.32%] vs D2D8 forelimbs n=17/88 [19.32%]; *p<0.005, z=2.35) and dramatic coordination disruptions on Sylgard (e, PD1 forelimbs n=8/189 [4.23%] vs D1D5 forelimbs n=23/177 [12.99%]; ***p<0.005, z=3.01; PD1 forelimbs n=8/189 [4.23%] vs D1D8 forelimbs n=33/171 [19.30%]; ***p<0.001, z=4.49; PD2 forelimbs n=3/81 [3.70%] vs D2D5 forelimbs n=34/76 [44.74%]; ****p<0.001, z=6.75; PD2 forelimbs n=3/81 [3.70%] vs D2D8 forelimbs n=24/80 [30.00%]; ****p<0.001, z=4.75).

The online version of this article includes the following source data and figure supplement(s) for figure 2:

**Source data 1.** File contains the raw data for *Figure 2* and *Figure 2—figure supplements 1 and 2*.

**Figure supplement 1.** Heterolateral demonstrates context-dependence, while homolateral limb pair relationships are context-independent during long descending propriospinal neuron (LDPN) silencing.

**Figure supplement 2.** Relationships between limb pairs are maintained during long ascending propriospinal neuron (LAPN) and long descending propriospinal neuron (LDPN) silencing.

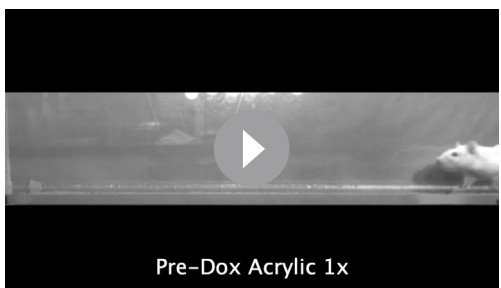

**Video 1.** Example of stepping during Pre-Dox on the Acrylic walking surface. Normal (1x) and one-quarter (.25x) speed.

https://elifesciences.org/articles/82944/figures#video1

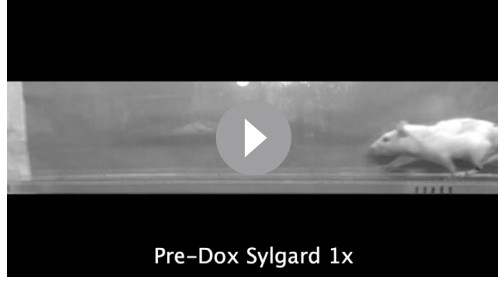

**Video 3.** Example of stepping during Pre-Dox on the Sylgard walking surface. Normal (1x) and one-quarter (.25x) speed.

https://elifesciences.org/articles/82944/figures#video3

the LDPN animals exhibited greater variability in many of the outcome measures, as compared to the LAPNs, but both groups showed a small, but statistically significant increase in mean speed during silencing regardless of stepping surface (*Figure 3K*). Thus, these findings suggest that both the LAPNs and LDPNs have roles in securing interlimb coordination but that removing them from the circuitry does not affect velocity-dependent step cycle characteristics or the locomotor rhythm in an otherwise intact rat.

## Silencing LDPNs preserves the stability of the stepping pattern despite disrupted coupling at the shoulder and hip girdles

In addition to stance, swing, and stride times, interlimb coordination involving the homolateral and heterolateral HL-FL pairs is a fundamental speed-dependent gait characteristic. These components of interlimb coordination were investigated by plotting the phases of each limb pair against each other at the control and Dox time points, and for each stepping surface. This comparison was not made previously for the LAPNs, so we include those results here (see *Pocratsky et al., 2020*; *Shepard et al., 2021* for LAPN methods). During control time points, steps were clustered in the center of the plot for the FL/FL and HL/HL comparisons, indicating that alternating forelimb steps were concomitant with alternating hindlimb steps (*Figure 2—figure supplement 1C and I*). Similarly, alternating hindlimb steps were clustered with synchronous/alternate forelimb steps, contralateral or ipsilateral, respectively (*Figure 2—figure supplement 1A, B, G and H*). This held true for control testing on both the low and high CoF surfaces. These phase plots demonstrate that during silencing the HL and FL pairs could adopt any possible phase value, but with some over-riding control system that maintained stable stepping even as the two limb pairs were partially decoupled. This can most easily be discerned by noticing that certain combinations of HL/HL – FL/FL phase values simply didn't occur (*Figure 2—figure supplement 1C, F, I and L*) even though each pair expressed the entire possible range of phase values. The spaces in these graphs, most notably centrally in D & J, and middle top and bottom in E & K, also support the concept that forelimb-hindlimb coordination

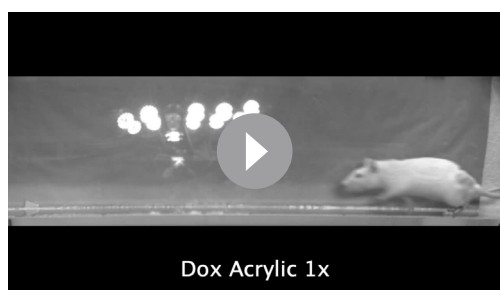

**Video 2.** Example of stepping during Dox on the Acrylic walking surface. Normal (1x) and one-quarter (.25x) speed.

https://elifesciences.org/articles/82944/figures#video2

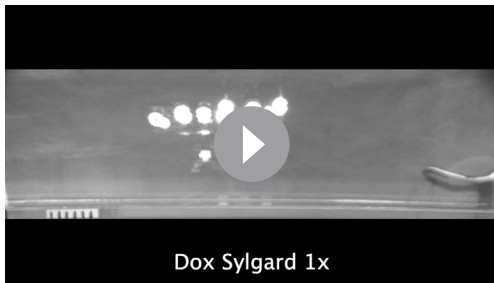

**Video 4.** Example of stepping during Dox on the Sylgard walking surface. Normal (1x) and one-quarter (.25x) speed.

https://elifesciences.org/articles/82944/figures#video4

**Table 1.** Watson's U² circular statistics calculations for hindlimb and forelimb pairs during long descending propriospinal neuron (LDPN) silencing on sylgard and acrylic.

We performed Watson's non-parametric two-sample U² circular statistics to determine function uncoupling in the hindlimb pair and forelimb pair. The null hypothesis tested was that two samples were from two populations with the same direction. Silencing the LDPNs only affected the directionality of the data on the Sylgard surface (Critical value of Watson's U² $_{(0.05,\infty,\infty)}$=0.1869; *Zar, 1974* Appendix D, Table D.44).

| Behavioral context | Watson's test | Left-right FL | Left-right HL |
|---|---|---|---|
| LDPN overground locomotion on Sylgard-coated surface | U² | 1.9012 | 2.0597 |
| | p-value | ****p<0.001 | ****p<0.001 |
| LDPN overground locomotion on acrylic surface | U² | 0.1549 | –0.5407 |
| | p-value | n.s. | n.s. |
| LDPN overground locomotion, Sylgard-coated vs acrylic (comparing control time points) | U² | 0.1689 | 0.0599 |
| | p-value | n.s. | n.s. |
| LDPN overground locomotion, Sylgard coated vs acrylic (comparing Dox$^{ON}$ time points) | U² | 0.3649 | 0.7102 |
| | p-value | ***p<0.005 | ****p<0.001 |

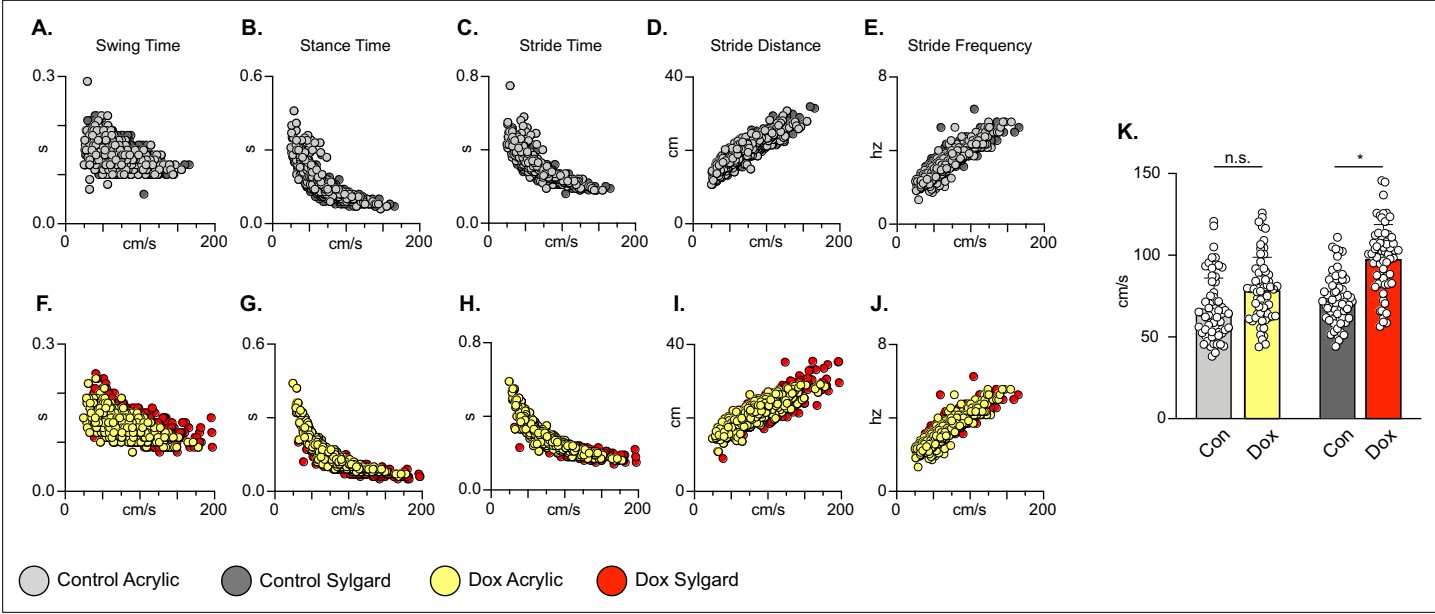

**Figure 3.** Silencing long descending propriospinal neurons (LDPNs) does not affect key features of locomotion. Spatiotemporal measures (swing time, stance time, stride time, stride distance) were plotted against speed for Control (**A–E**) and doxycycline (Dox) (**F–J**) time points. Lines of best fit are not displayed on graphs for clarity. Stance time and stride time display similar decay relationships during silencing of both pathways, regardless of surface (acrylic stance time: Control $R^2$=0.809 vs Dox $R^2$=0.896; Sylgard stance time: Control $R^2$=0.764 vs Dox $R^2$=0.811; LDPN Sylgard stride time: Control $R^2$=0.840 vs Dox $R^2$=0.842), while linear relationships are indicated for stride distance (**D**, **I**; acrylic stride distance: Control $R^2$=0.799 vs Dox $R^2$=0.745; Sylgard stride distance: Control $R^2$=0.780 vs Dox $R^2$=0.719) and stride frequency (**E**, **J**; acrylic stride frequency: Control $R^2$=0.864 vs Dox $R^2$=0.879; Sylgard stride frequency: Control $R^2$=0.793 vs Dox $R^2$=0.788). The average instantaneous speed was increased by Dox, regardless of walking surface (**K**; Average speed acrylic Control 67.09±18.97 vs Dox: 78.89±19.82, n.s.; Mixed-Model ANOVA; Average speed Sylgard Control 73.15±15.05 vs Dox: 98.13±20.63, ****p<0.05, Mixed-Model ANOVA).

The online version of this article includes the following source data for figure 3:

**Source data 1.** File contains the raw data for *Figure 3* showing the spatiotemporal characteristics of speed-dependent change in swing time, stance time, stride time, stride distance, and stride frequency for animals stepping on the acrylic and sylgard surfaces with and without Dox.

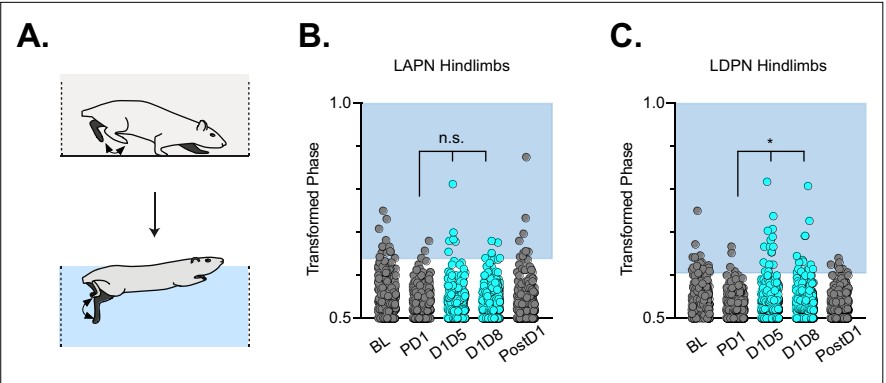

**Figure 4.** Hindlimb coordination during swimming is largely maintained during long ascending propriospinal neuron (LAPN) and long descending propriospinal neuron (LDPN) silencing. Similar to hindlimb coordination during overground locomotion, phase can be calculated for the R & L hindlimbs during a swimming task (**A**) Hindlimb alternation was maintained during silencing of LAPNs (**B**, PD1 hindlimbs n=4/294 [1.36%] vs D1D5 hindlimbs n=8/302 [2.65%]; n.s., z=1.13; PD1 hindlimbs n=4/294 [1.36%] vs D1D8 hindlimbs n=8/278 [2.88%]; n.s., z=1.25). However, silencing the LDPNs resulted in a small but statistically significant increase in the number of swim cycles with abnormal R-L phase (**C**, PD1 hindlimbs n=9/408 [2.21%] vs D1D5 hindlimbs n=23/416 [5.53%]; *p<0.05, z=2.5; PD1 hindlimbs n=9/408 [2.21%] vs D1D8 hindlimbs n=24/468 [5.13%]; *p<0.05, z=2.33).

The online version of this article includes the following source data for figure 4:

**Source data 1.** File contains the raw data for *Figure 4* showing the hindlimb-hindlimb (HL-HL) phase during swimming with the long ascending propriospinal neurons (LAPNs) silenced (from *Shepard et al., 2021*) or with the long descending propriospinal neurons (LDPNs) silenced (DOX^ON).

was only disrupted when the forelimb and hindlimb pairs were partially de-coupled and thus was a secondary rather than primary effect of silencing. In addition, there are several specific regions in these phase-phase plots that highlight the differences between the silenced phenotypes for the LAPNs and LDPNs. First, for the LAPNs, there were no perturbations in left-right alternation on the low CoF surface as illustrated by the closely clustered clouds of yellow markers in *Figure 2—figure supplement 1A, E and F*. In contrast, *Figure 2—figure supplement 1J, K and I* demonstrate that phase was disrupted for both walking surfaces when the LDPNs were silenced. Second, the appearance of steps with HL/HL and FL/FL phases ~0.25 & 0.75 is clear in *Figure 2—figure supplement 1F* as compared to 1 l, given the smaller number of steps analyzed for the LAPN animals. This is also reflected by comparing *Figure 2—figure supplement 1E and K*. In k, it is apparent that the ipsilateral HL-FL pair maintained a phase value around 0.5 even while the HL-HL phase varied throughout the entire range of values on both walking surfaces. Finally, no differences in the base-of-support were detected between the low and high CoF surfaces during silencing for either pathway, suggesting that balance/postural changes likely do not account for these fascinating results (LAPN Dox: 18.36±2.97° vs 21.44±4.48°, p>0.05, n.s., paired t-test; LDPN Dox: 18.95±7.41 vs 20.11±11.91, p>0.05, n.s., paired t-test).

## Interlimb coordination persists during swimming in intact animals

Finally, we examined the effects of LDPN silencing on left-right hindlimb coordination in water (*Figure 4A*). Swimming is a bipedal task where the hindlimbs provide the major propulsive force while the forelimbs steer (*Gruner and Altman, 1980*). During swimming, the limbs are unloaded and the proprioceptive and cutaneous feedback associated with stepping is altered (*Miller and van der Burg, 1973*; *Duysens and Stein, 1978*; *Akay et al., 2014*). In contrast to our overground findings, and in keeping with our previous LAPN study (*Pocratsky et al., 2020*), silencing the LDPNs had no effect on left-right hindlimb alternation during swimming (*Figure 4B and C*). This observation suggests that the LDPN locomotor phenotype is not rigidly context-independent. Thus, the circuitry responsible for securing alternation during swimming remains unperturbed when either set of long propriospinal neuron is silenced.

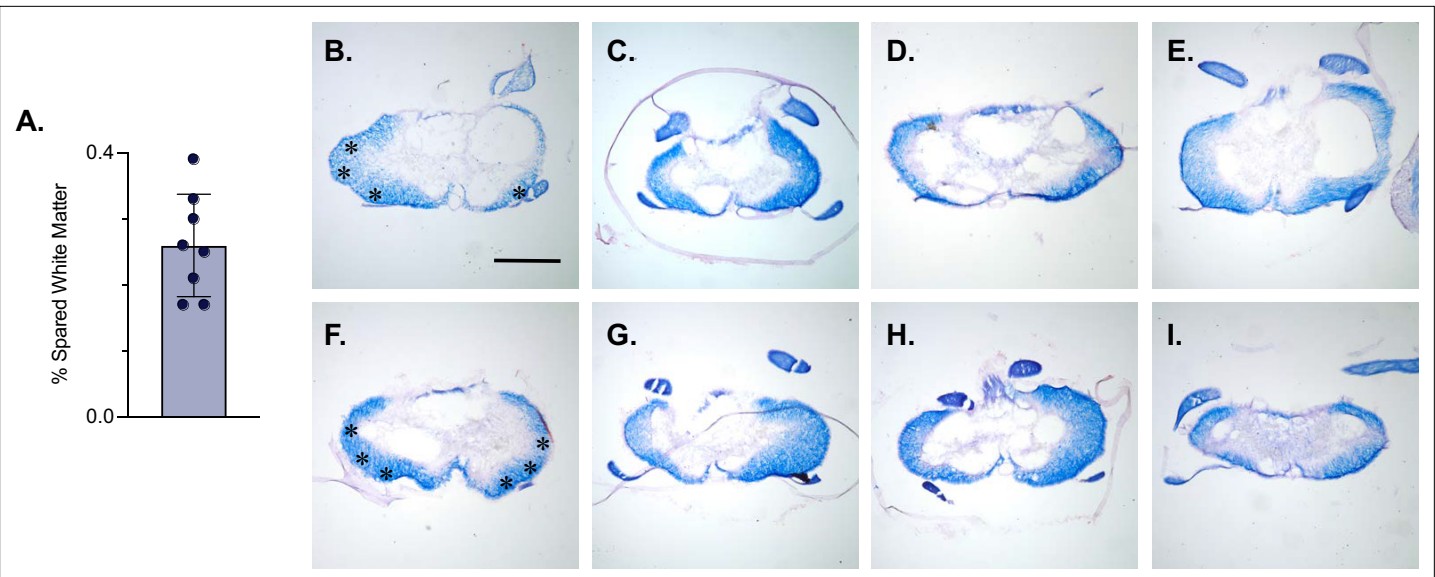

**Figure 5.** White matter sparing is present at the injury epicenter. Spared white matter percentage at the injury epicenter ranges from 17–39% (**A**). White matter damage at the spinal cord injury epicenter as confirmed by histology (**B–I**). Individual images represent the injury epicenter of each animal used in the main dataset (n=8; average white matter percentage: 26%, SD 7.76%). Asterix in (**B**) and (**F**) indicate regions where spared LAPN axons would exist. Scale bar = 1 mm.

The online version of this article includes the following source data for figure 5:

**Source data 1.** File contains the raw data for *Figure 5* showing the percent spared white matter at the injury epicenter derived from eriochrome cyanin stained section.

## Presence of putatively eTeNT-positive synapses in the lumbar spinal cord and cell bodies in the cervical spinal cord after spinal cord injury

LDPNs with cell bodies in the cervical enlargement and axon terminals in the more rostral segments of the lumbar cord have axons that travel in the ventrolateral funiculus (VLF) (*Menétrey et al., 1985*; *Reed et al., 2006*; *Flynn et al., 2017*). Lesions to the thoracic VLF in the cat disrupt forelimb–hindlimb coupling (*Brustein and Rossignol, 1998*), further confirming the location of these long descending propriospinal projections. Given this location, it was essential to determine whether the VLF, the area of white matter in which LDPN axons project, was preserved after SCI. To examine this, animals were euthanized during post-injury Dox administration. Their spinal cords were harvested and were blocked in three sections: caudal cervical spinal cord (cell bodies), caudal thoracic spinal cord (injury epicenter and penumbra), and rostral lumbar spinal cord (cell terminals). Thoracic tissue sections were stained at the injury epicenter using eriochrome cyanine (EC) to quantify to amount of spared white matter and imaged to determine where the spared white matter was located. Amongst all animals, the average spared white matter percentage (SWM%) was 26%, slightly higher than that previously described in the LAPN SCI study (*Figure 5A*). The proposed location of LDPN axons within the VLF was spared, as evidenced by the remaining spared white matter in each animal (*Figure 5B–I*). Thus, a proportion of LDPN axons are likely spared after SCI, a critical finding for the interpretation of any behavioral assessments moving forward.

Knowing that LDPN axons were likely preserved at the level of injury, we next confirmed that eTeNT.EGFP-expressing putative LDPN axons were present in the rostral lumbar spinal cord, as well as eTeNT.EGFP-expressing putative LDPN cell bodies in the intermediate gray matter of the caudal cervical segments. Histological analysis for EGFP immunoreactivity showed that putatively positive eTeNT.EGFP fibers were found to surround neuronal processes in the rostral lumbar enlargement (*Figure 6A*). eTeNT.EGFP co-localized with synaptophysin (*Figure 6B*; synaptic marker), vesicular GABA transporter (*Figure 6D*, VGAT, inhibitory neurotransmitter), and vesicular glutamate transporter 1/2 (*Figure 6E*; VGlut2, excitatory neurotransmitters). For D and E, arrows indicate two neuron terminals shown in X, Z and Y, Z focal planes in the boxes below and to the right of these photomicrographs. Isotype controls revealed minimal-to-no immunoreactivity (*Figure 6C*).

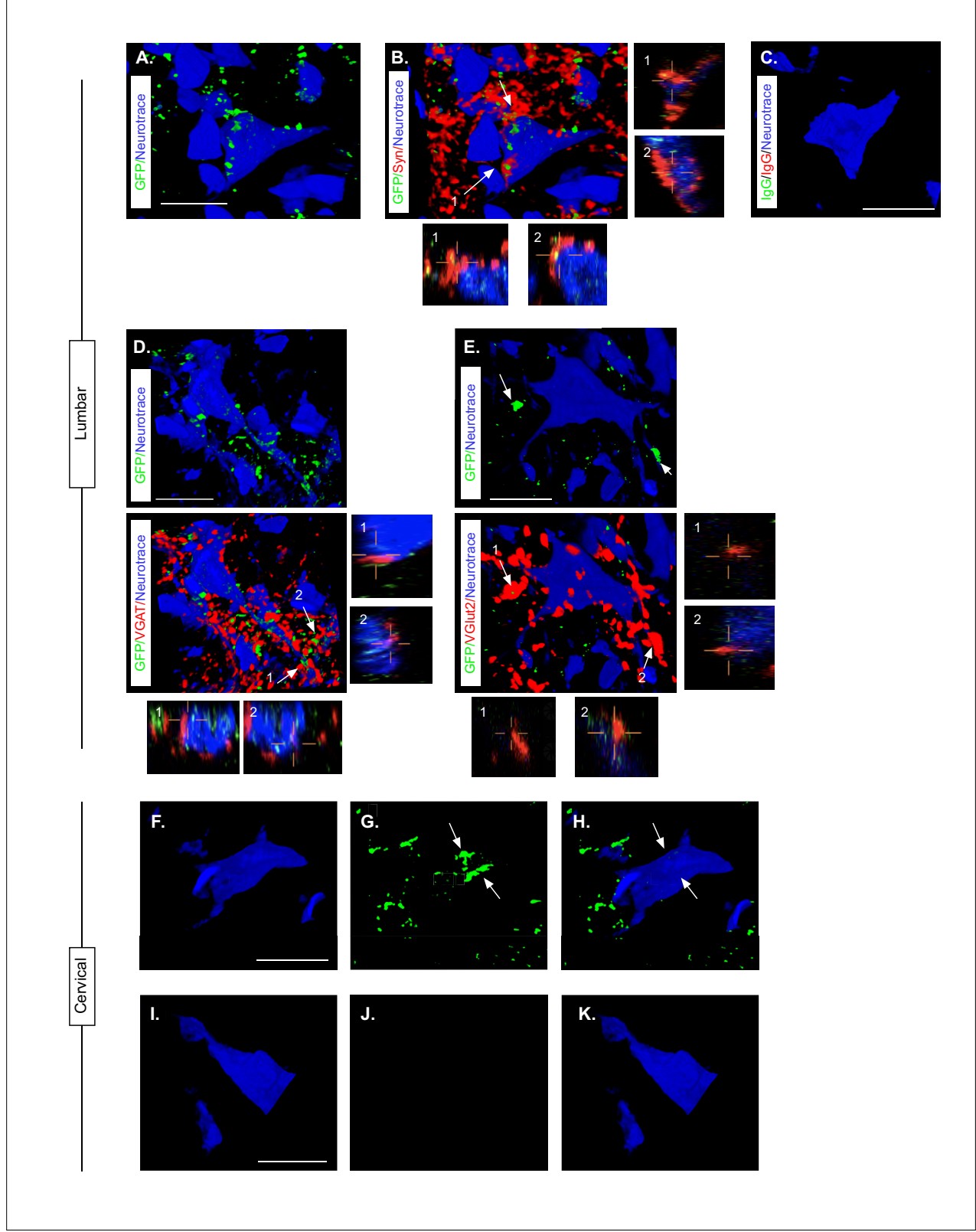

**Figure 6.** Presence of eTeNT-EGFP in putatively silenced long descending propriospinal neurons (LDPNs) across the level of injury. High magnification, volume-rendered images demonstrating eTeNT.EGFP putatively positive fibers (green) surrounding fluorescent Nissl-stained neuronal cell bodies (Neurotrace - blue) and synaptophysin-labeled putative synapses (red) in lumbar spinal cord segments of interest (**A, B**, 100 x magnification, L1-L2 spinal cord). White arrows indicate areas of colocalization, which are further marked by numbers and xz, yz planes. Isotype control reveals minimal

*Figure 6 continued on next page*

*Figure 6 continued*

immunoreactivity (**C**, IgG controls for eTeNT.EGFP shown). eTeNT.EGFP (green) signal co-localizes with neuronal cell bodies/processes (blue) and with inhibitory neurotransmitter marker vesicular GABA transporter (**D**, VGAT, red), and excitatory neurotransmitter vesicular glutamate transporter 2 (**E**, VGlut2, red). eTeNT.EGFP putative cell bodies (green) in the cervical spinal cord colocalized with fluorescent nissl stained neurons (Neurotrace - blue) (**F-H**, C6-C7 spinal cord). Minimal presence of eTeNT.EGFP signal in isotype controls (**I-K**, C6-C7 spinal cord). Scale bars = 20 μm.

Using the same immunohistological protocol as described for lumbar spinal tissue, caudal cervical spinal cord segments were assessed for the presence of eTeNT.EGFP within LDPN somata. Putatively eTeNT-EGFP-positive LDPN cell bodies co-localized with fluorescent Nissl stained (NeuroTrace) neurons in the intermediate gray matter (*Figure 6f–H*, *Reed et al., 2006*). Isotype controls showed no immunoreactivity (*Figure 6I–K*). Taken together with the lumbar spinal cord histology, these data suggest that double-infected LDPNs maintained expression of eTeNT at the level of both the cell soma and axon terminals following SCI. Furthermore, any post-SCI spared LDPNs axons were functionally silenced, indicating that any behavioral changes seen during Dox administration were concomitant with active eTeNT.EGFP expression.

## Silencing LDPNs post-SCI restores coordination indices and improves gross locomotor outcomes

Having validated that (1) the viral-based silencing system is active post-SCI and (2) LDPN axons are likely intact post-SCI, we began to explore the effects of LDPN silencing on locomotion after injury. In our previous study of the LAPNs, we found that moderate contusive injuries at T10 resulted in disruptions of hindlimb alternation that were not dissimilar to the effects of LAPN silencing. However, silencing LAPNs post-SCI unexpectedly resulted in improved stepping, and in particular restored hindlimb alternation so that it was not significantly different from the uninjured state (*Shepard et al., 2021*). Thus, given that LDPN silencing also resulted in disrupted right-left alternation, we hypothesized that silencing LDPNs after a T10 SCI would again result in improved recovered locomotion.

The Basso, Beattie, Bresnahan (BBB) Locomotor Rating Scale was used to grossly evaluate locomotor recovery (*Basso et al., 2002*). Control BBB scores (post-injury, pre-silencing) were concentrated around a score of 12, consistent with previous literature for this injury severity and with our previous LAPN silencing study (*Shepard et al., 2021*; *Basso et al., 2002*; *Smith et al., 2006*). Analogous to our LAPN study, average BBB scores were modestly increased during silencing (*Figure 7A*), with a greater proportion of raw scores for each limb between 13–18 (*Figure 7B*). The increased BBB scores were due to improved weight support of the hindlimbs as well as better 'coordination,' resulting in the appearance of improved stepping during silencing. Some BBB scores remained around 13, indicating that not all animals' stepping improved at all Dox time points. However, a significant proportion of BBB scores were increased, demonstrating that better stepping occurred in multiple animals at multiple time points as a result of silencing.

To further assess the overall locomotor function post-SCI and the impact of LDPN silencing, we evaluated the step sequence patterns using the regularity index (RI) and coordinated pattern index (CPI). As previously described, RI scores plantar steps according to paw placement order and represents the gait as the ratio of plantar steps that are in order (one of four different orders that normal rats exhibit) over the total number of steps, while CPI represents the number of correctly patterned step cycles (dorsal and plantar) to the total number of cycles (dorsal and plantar) (*Hamers et al., 2001*; *Caudle et al., 2015*). CPI accounts for whether the animal can achieve coordination regardless of their ability to attain plantar placement, while RI only accounts for correctly patterned plantar steps. RI and CPI were both modestly improved during Dox administration (*Figure 7C and D*). We also examined the plantar stepping index (PSI), an index that simply compares the number of hindlimb plantar steps as compared to forelimb plantar steps, and it was also modestly improved during silencing (*Figure 7E*).

When using RI and CPI, an important differentiator is the presence of dorsal steps, which occur when the walking surface is contacted by the dorsum of the foot and/or toes because they remain flexed or curled at stance. Dorsal steps are common after SCI, especially acutely, and are one important measure of recovery from low-thoracic contusive SCI (*Basso et al., 1996*). Dorsal stepping index (DSI) is simply the ratio of dorsal steps to the number of total hindlimb steps, thus the DSI in normal uninjured animals should be zero. We compared DSI during post-injury control and silenced time points. The DSI was significantly reduced from 24.31±21.02–9.64±8.70 during silencing indicating a

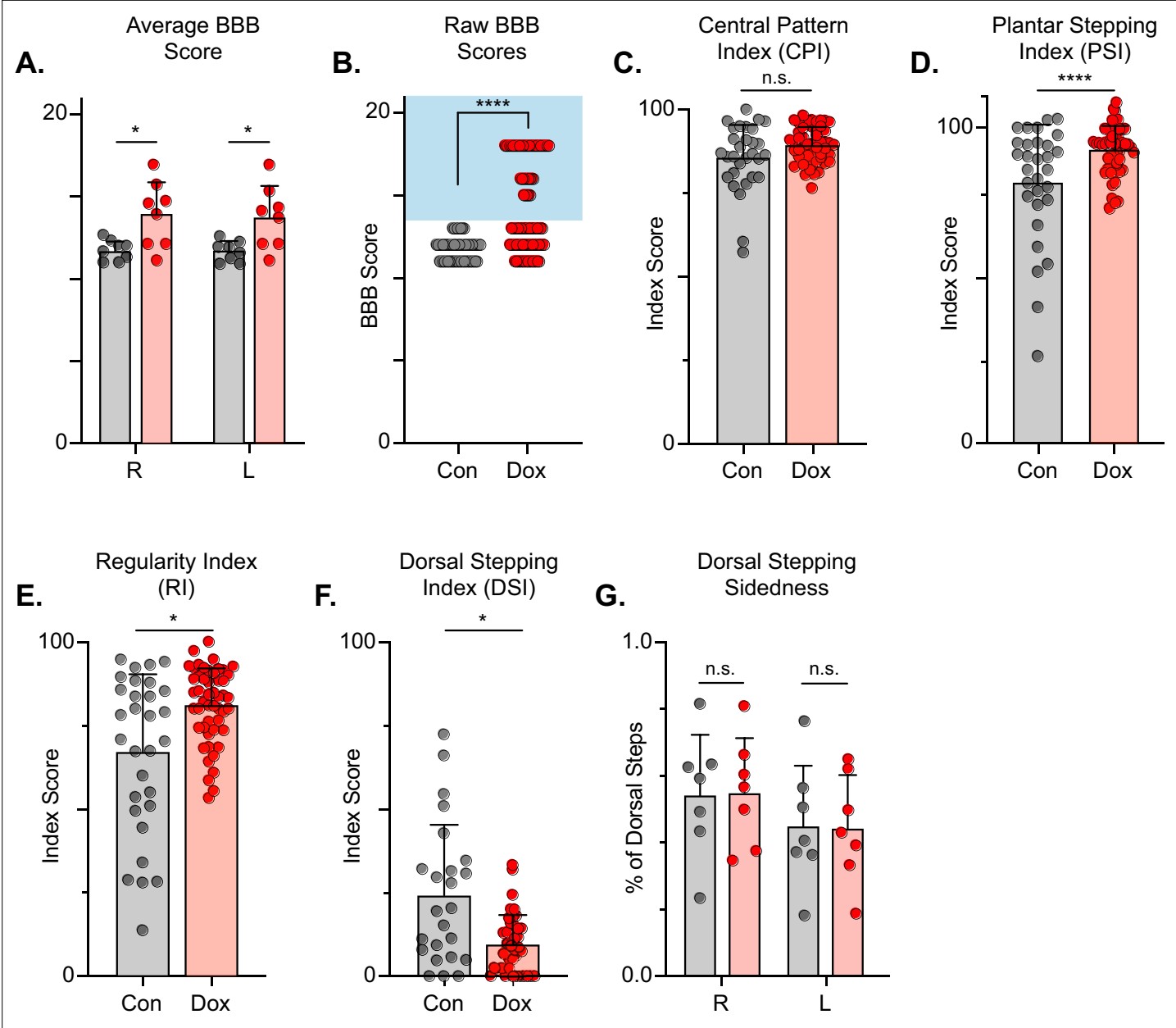

**Figure 7.** Silencing long descending propriospinal neurons (LDPNs) post-spinal cord injury (SCI) restores some coordination indices and improves gross locomotor outcomes. Average Basso, Beattie, Bresnahan (BBB) scores for Control and doxycycline (Dox) time points are shown for each hindlimb (**A**, group average ± SD; Control Left to Dox Left, *p=0.021; Control Right to Dox Right, n.s. p=0.019; mixed model ANOVA, Bonferroni *post hoc*). No significant difference was found between right and left BBB scores so they were combined for average and raw scores (mixed model ANOVA, Bonferroni *post hoc*; statistics not shown on graph). Right and left hindlimb raw BBB scores are shown in (**B**). (Control: n=0/48 [0.0%] vs Dox: n=28/80 [35.0%]; ****p<0.001, z=4.64; Binomial Proportion Test; circles = individual left or right BBB scores; shaded region = values beyond control variability). Average coordinated pattern index (CPI) scores were not different (**C**, Control CPI: 84.47±8.61 vs Dox CPI: 88.28±3.07, t=1.66, df = 7, n.s. p=0.141; paired t-test). Average plantar stepping index (PSI) scores were significantly different (**D**, Control PSI: 79.32±15.97 vs Dox PSI: 91.71±6.45, t=3.22, df = 7, *p=0.015; paired t-test). Average regularity index (RI) scores were significantly different (**E**, Control RI: 60.92±3.61 vs Dox RI: 78.81±8.97, t=7.69, df = 7, ****p>0.001; paired t-test). as was the average dorsal stepping index (DSI) scores (**F**, Control DSI: 24.31±18.49 vs Dox DSI: 9.64±6.66, t=3.33, df = 7, *p=0.013; paired t-test). Data are shown with individual animal scores from each Control and Dox time point to show variability (gray circles and red circles for Control and Dox, respectively). Dorsal stepping index accounts for total dorsal steps for both left and right hindlimbs. The total dorsal steps were separated based on sidedness: right hindlimb (RHL) and left hindlimb (LHL) (**G**, Control right: 0.546±0.183 vs Dox right: 0.553±0.162, df = 6, n.s. p=0.942; Control left: 0.454±0.183 vs Dox left: 0.447±0.162, df = 6, n.s. p=0.942; mixed model ANOVA). The right hindlimb showed more dorsal steps overall; however, it maintained that percentage during Dox. No significant differences were seen in sidedness (mixed model ANOVA, statistics not shown on graph).

*Figure 7 continued on next page*

*Figure 7 continued*

The online version of this article includes the following source data for figure 7:

**Source data 1.** File contains the raw data for *Figure 7* showing the gross motor scores including the Basso, Beattie, Bresnahan (BBB) Open Field Locomotor Scale scores and subscores, the Central Pattern, Regularity, Plantar Stepping, Dorsal Stepping, and Dorsal Sidedness indices.

significant reduction in the overall proportion of dorsal steps (*Figure 7F*). This decrease closely resembles the decrease seen in dorsal stepping indices during LAPN post-SCI silencing.

We also compared the sidedness of dorsal steps. Dorsal step-sidedness refers to the percentage of dorsal steps that occur on the right and the percentage that occurs on the left. The sidedness did not change between control and silencing, with ~55% of dorsal steps occurring in the right hindlimb and ~45% of dorsal steps occurring in the left hindlimb during both control and silenced time points (*Figure 7G*). This is an important observation, as it shows that one limb did not improve to a greater extent than the other. For the purposes of simplicity and ease of comparison with pre-injury data, we utilize the left limb as lead to demonstrate the principle behavioral changes post-SCI, as we did with LAPN SCI data. To further elucidate these improvements in coordination indices, we examined more specific intralimb kinematics and interlimb gait analyses.

## LDPN silencing leads to negligible improvements in intralimb coordination after SCI

We next examined intralimb coordination using the peak-trough excursions and the temporal relationship between the proximal (iliac crest – hip – ankle) and distal (hip – ankle – toe) angles of the three-segment, two-angle kinematic hindlimb model we employ (*Figure 8A–D*; *Videos 5 and 6*). Interestingly, the excursions of both the distal and proximal angles were not significantly improved during silencing (*Figure 8E and F*). Next, we explored the temporal relationship between the proximal and distal angles. In uninjured animals, the peak extension of the distal angle typically occurs almost in-phase with the peak extension of the proximal angle during the stance phase of the step cycle, resulting in a coordination value ranging from 0.9 to 0.1 on a linear scale. Following injury, the average coordination value of these two angles was unchanged between Control and Dox, even while the variability was somewhat reduced (*Figure 8G*), indicating that silencing the LDPNs had a negligible effect on intralimb coordination post-SCI.

## Hindlimb-hindlimb, but not hindlimb-forelimb, coupling relationships were restored during post-SCI silencing

After SCI, the temporal relationship between hindlimb and hindlimb-forelimb limb pairs (HL-HL and HL-FL) is highly variable, even as the forelimbs maintain alternation. This variability is accounted for, in part, by the presence of dorsal steps. However, many plantar steps also fall into irregular gait patterns after SCI. We included both plantar and dorsal steps for the purposes of quantifying interlimb coordination, with left dorsal steps identified as teal throughout the remaining figures. As was seen during LAPN silencing, the hindlimb-hindlimb pair and the hindlimb-forelimb pairs are partially decoupled at all post-SCI Control time points (*Figure 9A*, gray circles). We found that the coupling of the hindlimbs was significantly restored during the initial round of post-injury silencing (D2D5 & D2D8). Heterolateral coupling (HL-FL) was also improved, but reached significance only on D8. Interestingly, the improvement in HL-HL coupling was not significant at any Dox3 time point, suggesting that there may be some plastic changes occurring within the circuitry over this 3 week period from D2D5 to D3D14. Importantly, hindlimb coupling is seen as significantly improved when Control and Dox time points are collapsed, suggesting that the total number of steps that fall outside normal variability are reduced overall by LDPN silencing from ~23 – ~12% (*Figure 9A*). In addition, the coupling of the heterolateral hindlimb-forelimb pair was also significantly improved overall, when the control and Dox time points were collapsed (*Figure 9B*), even though the absolute reduction in abnormal steps was modest (from ~43 – ~37%; *Figure 9B*). Finally, forelimb-forelimb alternation was maintained during both control and Dox time points (*Figure 9C*).

Two animals were removed from the dataset prior to injury as they showed no perturbations to left-right alternation at any pre-injury Dox time point (*Figure 9—figure supplement 1A-D*, see Methods for exclusion criteria). Interestingly, these animals showed an injury-induced disruption and did not

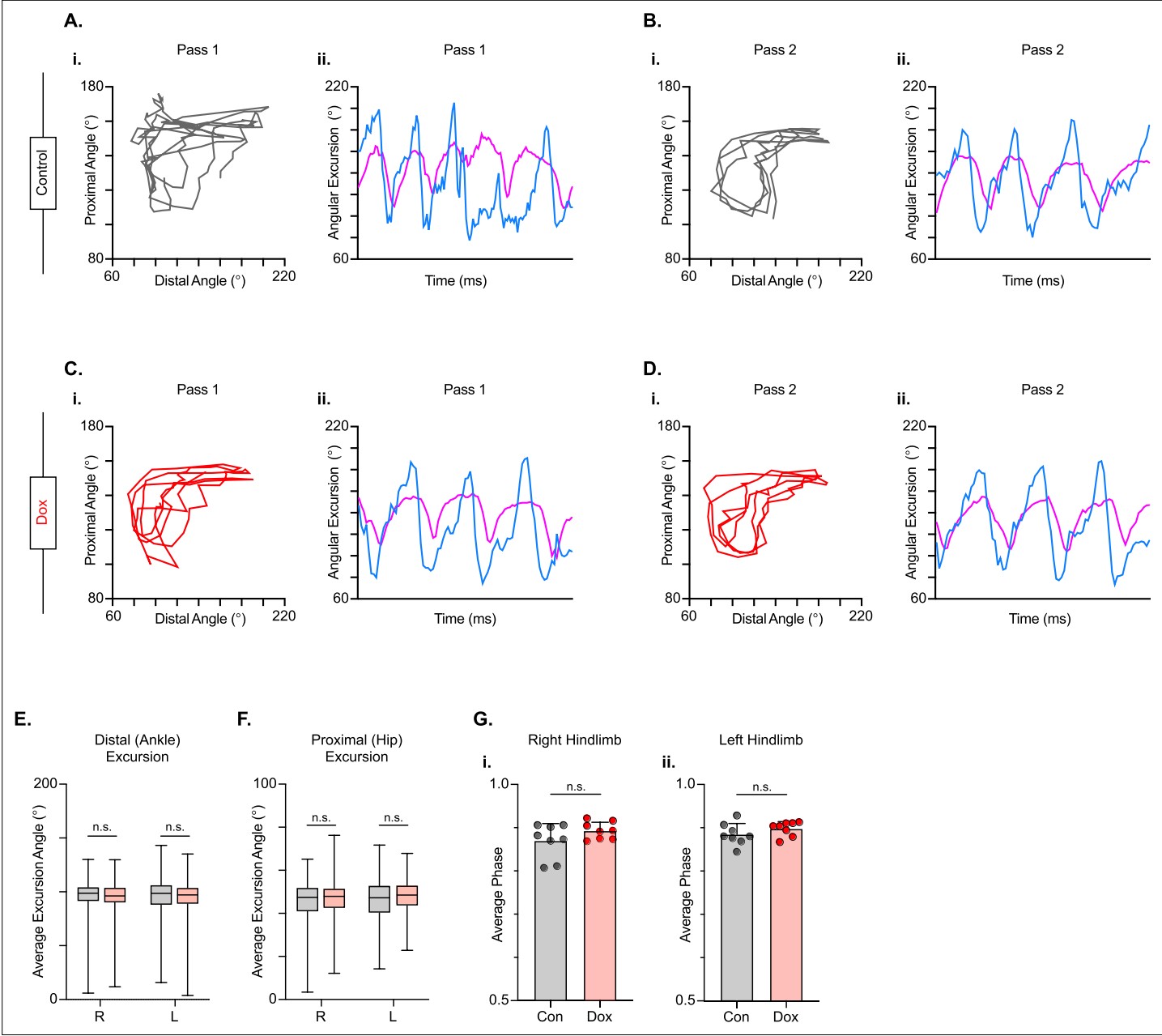

**Figure 8.** Long descending propriospinal neuron (LDPN) silencing has no effect on intralimb coordination after spinal cord injury. Angle-angle plots and excursion traces are shown using our three-segment, two-angle model of the hindlimb which incorporates the hip-knee into the proximal angle and the knee-ankle into the distal angle. Shown are the proximal (purple) and distal (blue) angles during control (**A, B**) and doxycycline (Dox) (**C, D**) time points. Distal angle excursion (**E**, Control right average: 90.90 vs Dox right average: 93.45, df = 7, n.s., R.M. ANOVA; Control left average: 91.49 vs Dox left average: 93.81, df = 7, n.s., R.M. ANOVA) and proximal angle excursion (**H**, Control right average: 42.67 vs Dox right average: 45.93, df = 7, n.s., R.M. ANOVA; Control left average: 45.08 vs Dox left average: 47.45, df = 7, n.s., R.M. ANOVA; middle bar indicates group average with extension bars indicating the range of raw data) were unchanged as a result of silencing. Intralimb phase values relating the peak of one angle to the peak of the other are converted from a scale of 0–1 to 0.5–1.0 to be viewed linearly (**G**). The average intralimb coordination values for the right and left hindlimbs are plotted for Control and Dox time points (Control right: 0.87±0.04 vs Dox right: 0.89±0.02, t=2.44, df = 7, n.s. p=0.045, paired t-test; Control left: 0.88±0.03 vs Dox left: 0.90±0.02; t=1.16, df = 7, n.s. p=0.286, paired t-test; circles = individual step cycles; shaded region = values beyond control variability). No significant differences in intralimb coordination were found.

The online version of this article includes the following source data for figure 8:

**Source data 1.** File contains the raw data for *Figure 8* showing the kinematic measures of hip-ankle-toe (HAT), iliac crest-hip-ankle (IHA), the excursion of each, and the phase difference for these two angles.

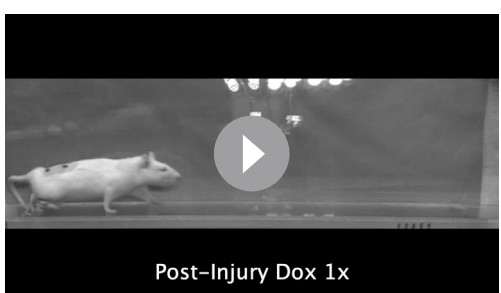

**Video 5.** Example of post-injury stepping during Dox on the Sylgard walking surface. Normal (1x) and one-quarter (.25x) speed.
https://elifesciences.org/articles/82944/figures#video5

show improvements in coordinated stepping during LDPN silencing post-SCI, suggesting that the pre- and post-SCI changes are due to LDPN silencing (*Figure 9—figure supplement 1*).

To summarize the findings on interlimb coordination, post-injury silencing significantly improved hindlimb-hindlimb alternation, had only a modest effect on the hindlimb-forelimb relationship and no impact on right-left forelimb coordination, which remained alternating throughout after injury. Thus, the injury eliminated the impact of LDPN silencing on forelimb circuitry even though the cell bodies of the neurons being silenced are rostral to the injury. In contrast, in our previous work, we showed that silencing the LAPNs after injury improved the interlimb coordination (alternation) of the hindlimbs with essentially no impact on homo- or heterolateral interlimb (HL-FL) coupling (*Shepard et al., 2021*). Finally, the reduced prevalence of dorsal steps during post-injury silencing of both LAPNs and LDPNs accounted for some, but not all, of the improvement in hindlimb interlimb coordination.

## Silencing LDPNs modestly improves balance and posture post-SCI

Populations of descending cervico-lumbar projections have been implicated as essential for postural stability during overground locomotion in the uninjured mouse (*Ruder et al., 2016*). To determine if this holds true for the LDPNs post-SCI, we examined measures that are associated with balance and postural control. Uninjured animals typically have a relatively narrow base of support. In conditions with increased postural instability, such as SCI, the paws become externally rotated and the base of support widens (*Basso et al., 1996*). Thus, we looked at gait angle, the angle made by three subsequent hindlimb placement points (R-L-R), diagonal length (distance from L hindpaw to R forepaw placement), and rear track width (distance between R and L hindpaw placements) and found that all three were modestly improved during silencing (*Figure 10A–C*).

We next challenged the animals' ability to maintain balance and posture by testing them on a horizontal ladder task (*Figure 10D*). To successfully traverse the ladder apparatus, animals must maintain postural control to accurately place their feet on the fixed-space ladder rungs. The number of foot slips, or times that the foot doesn't maintain contact with a rung, was reduced for both the left and right hindlimbs (*Figure 10D*) during post-injury LDPN silencing.

## Key features of locomotion are restored during post-SCI LAPN silencing

In the uninjured animal, silencing did not affect the fundamental relationships between speed and spatiotemporal features of limb movements (e.g. swing time, stance time, etc.). However, these relationships are disrupted by the SCI model chosen, such that a substantial proportion of dorsal and some plantar steps fall outside the typically described relationship (*Figure 11A-E*). Silencing LDPNs restored these relationships and resulted in a dramatic reduction in the variability of abnormal steps, regardless of dorsal or plantar stepping (*Figure 11F–J*). Importantly, average swing (*Figure 11K*) and stance times (*Figure 11L*) were unchanged during post-SCI silencing as was the duty cycle (*Figure 11M*). Finally, the average locomotor speed was modestly increased during post-injury silencing (*Figure 11N*) suggesting that speed may be a factor in the improved

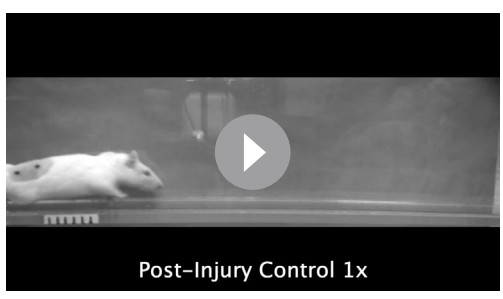

**Video 6.** Example of post-injury stepping during Dox-off (control) on the Sylgard walking surface. Normal (1x) and one-quarter (.25x) speed.
https://elifesciences.org/articles/82944/figures#video6

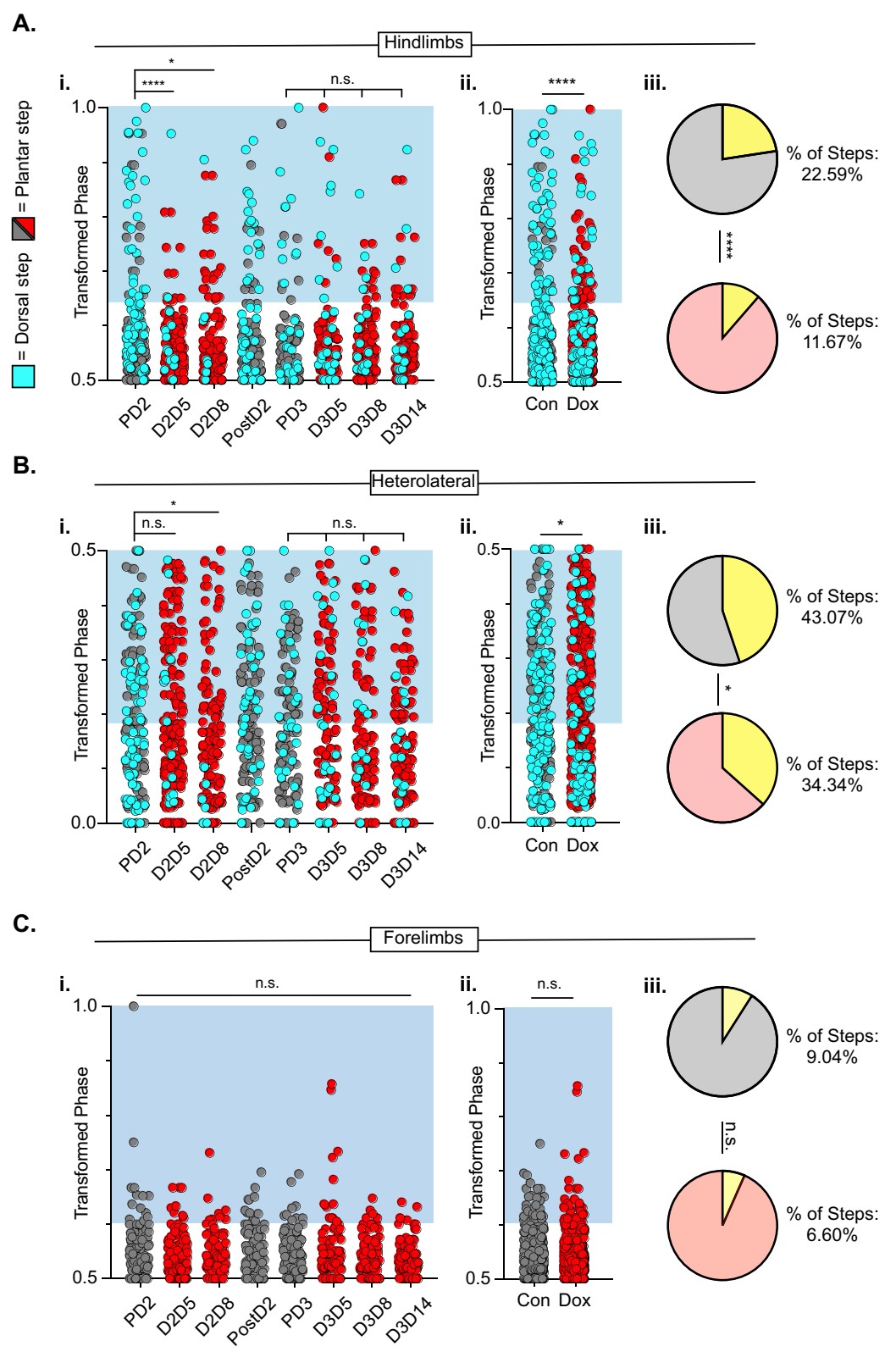

**Figure 9.** Hindlimb-hindlimb and hindlimb-forelimb coupling relationships are marginally improved while forelimb-forelimb coupling is unchanged during post-spinal cord injury (SCI_ long descending propriospinal neuron LDPN) silencing. Transformed phase values for hindlimbs at each post-injury time point are separated in **A**ᵢ and are collapsed into Control and Dox in **A**ᵢᵢ. # steps beyond control variability: PD2 hindlimbs n=33/114 [28.94%]

*Figure 9 continued on next page*

*Figure 9 continued*

vs D2D5 hindlimbs n=8/107 [7.48%]; ****p<0.001, z=4.10; PD2 hindlimbs n=33/114 [28.94%] vs D2D8 hindlimbs n=17/108 [15.74%]; *p<0.05, z=2.35; PD3 hindlimbs n=18/107 [16.82%] vs D3D5 hindlimbs n=13/108 [12.03%]; n.s., z=0.97; PD3 hindlimbs n=18/107 [16.82%] vs D3D8 hindlimbs n=12/100 [12%]; n.s., z=1.04; PD3 hindlimbs n=18/107 [16.82%] vs D3D13 hindlimbs n=12/107 [11.21%]; n.s., z=1.18. Control hindlimbs with dorsal steps: n=75/332 [22.59 %] vs Dox hindlimbs with dorsal steps: n=62/530 [11.70 %]; ****p<0.001, z=4.23, B.P. test. Transformed phase values are also shown for the heterolateral hindlimb-forelimb pairs are shown in (**B**$_i$), Dorsal steps were similarly collapsed into Control and doxycycline (Dox) in **B**$_{ii}$. # steps beyond control variability: PD2 heterolateral n=47/114 [41.23%] vs D2D5 heterolateral n=42/107 [39.25%]; n.s., z=0.3; PD2 heterolateral n=47/114 [41.23%] vs D2D8 heterolateral n=28/108 [25.92%]; p<0.05, z=2.41; PD3 heterolateral n=44/107 [41.12%] vs D3D5 heterolateral n=49/108 [45.37%]; n.s., z=0.47; PD3 heterolateral n=44/107 [41.12%] vs D3D8 heterolateral n=37/100 [37%]; n.s., z=0.72; PD3 heterolateral n=44/107 [41.12%] vs D3D13 heterolateral n=36/107 [33.64%]; n.s., z=1.13, B.P. tests. Control heterolateral limbs with dorsal steps: n=143/332 [43.07 %] vs Dox heterolateral limbs with dorsal steps: n=192/530 [36.16 %]; *p<0.05., z=2.01, B.P. tests. Plantar steps are indicated by gray circles (Control) and red circles (Dox), while dorsal steps are indicated by teal circles for both Control and Dox datasets. The blue boxes indicate values outside of normal variability for the specified uninjured limb pair mean. The percentage of abnormal steps found above normal variability is calculated for their respective limb pairs (**A**$_{iii}$, **B**$_{iii}$; statistics as shown above, B.P test). Forelimb-forelimb coupling data is represented in **C**$_i$-**C**$_{iii}$. No dorsal steps are seen for forelimb coupling, therefore, all steps represented for forelimb graphs are plantar. Transformed phase values are shown for the forelimb-forelimb pair in (**C**$_i$) with all time points collapsed in (**C**$_{ii}$). # steps beyond control variability: PD2 forelimb n=12/114 [10.52%] vs D2D5 forelimb n=9/107 [8.41%]; n.s., z=0.54; PD2 forelimb n=12/114 [10.52%] vs D2D8 forelimb n=7/108 [6.48%]; n.s., z=1.08; PD3 forelimb n=8/107 [7.47%] vs D3D5 forelimb n=11/108 [10.19%]; n.s., z=0.7; PD3 forelimb n=8/107 [7.47%] vs D3D8 forelimb n=5/100 [5%]; n.s., z=0.73; PD3 forelimb n=8/107 [7.47%] vs D3D13 forelimb n=3/107 [2.80%]; n.s., z=1.55, B.P. tests. Control forelimb steps: n=30/332 [9.04%] vs Dox forelimb steps: n=35/530 [6.60%]; n.s., z=1.32, B.P. tests.

The online version of this article includes the following source data and figure supplement(s) for figure 9:

**Source data 1.** File contains the raw data for *Figure 9* showing the phase differences for hindlimbs, heterolateral limb pairs, and the forelimbs, including the percentage of dorsal steps.

**Figure supplement 1.** Animals excluded based on lack of behavioral outcomes pre-injury show no improvements in hindlimb coupling post-injury.

**Figure supplement 1—source data 1.** File contains the raw data for *Figure 9—figure supplement 1* which includes the phase data for the animals that were removed from the study because they did not show a Dox[ON] silenced phenotype.

spatiotemporal relationships. Together with the previous findings, these data suggest that silencing LDPNs after SCI positively influences the temporal coordination of multiple aspects of locomotion, primarily interlimb (HL-HL) coordination. It is imperative to note that while the pattern of improvements observed with LDPN silencing post-SCI is very similar to those seen when the LAPNs were silenced, the overall magnitude of improvement is modest by comparison, where the LAPN silencing induced a more robustly improved phenotype.

## Discussion

Considerable evidence suggests that LDPNs play a role in interlimb coordination in uninjured animals (*Matsushita et al., 1979*; *Skinner et al., 1980*; *Menétrey et al., 1985*; *Alstermark et al., 1987*; *Nathan et al., 1996*). Supporting those data, current results indicate that inter-enlargement LDPNs are an essential pathway in securing interlimb coordination of both the forelimb and hindlimb pairs. When LDPNs were removed from the circuitry via synaptic silencing, there was a greater impact on hindlimb alternation than on the forelimbs, as shown by the higher proportion of affected steps. Furthermore, disrupted forelimb-hindlimb coordination was observed only when the right-left alternation of the forelimb and hindlimb pairs occurred, suggesting that the latter is the primary impact of silencing while the former is a secondary effect. The role of LDPNs in locomotor function after SCI has not previously been reported. We found that silencing the LDPNs after SCI brought about modest but meaningful improvements in postural stability concomitant with mild but significant improvements in hindlimb-hindlimb coordination, including paw placement order and timing, and speed-dependent gait indices. These improvements occurred in the absence of any disruption to forelimb alternation, a

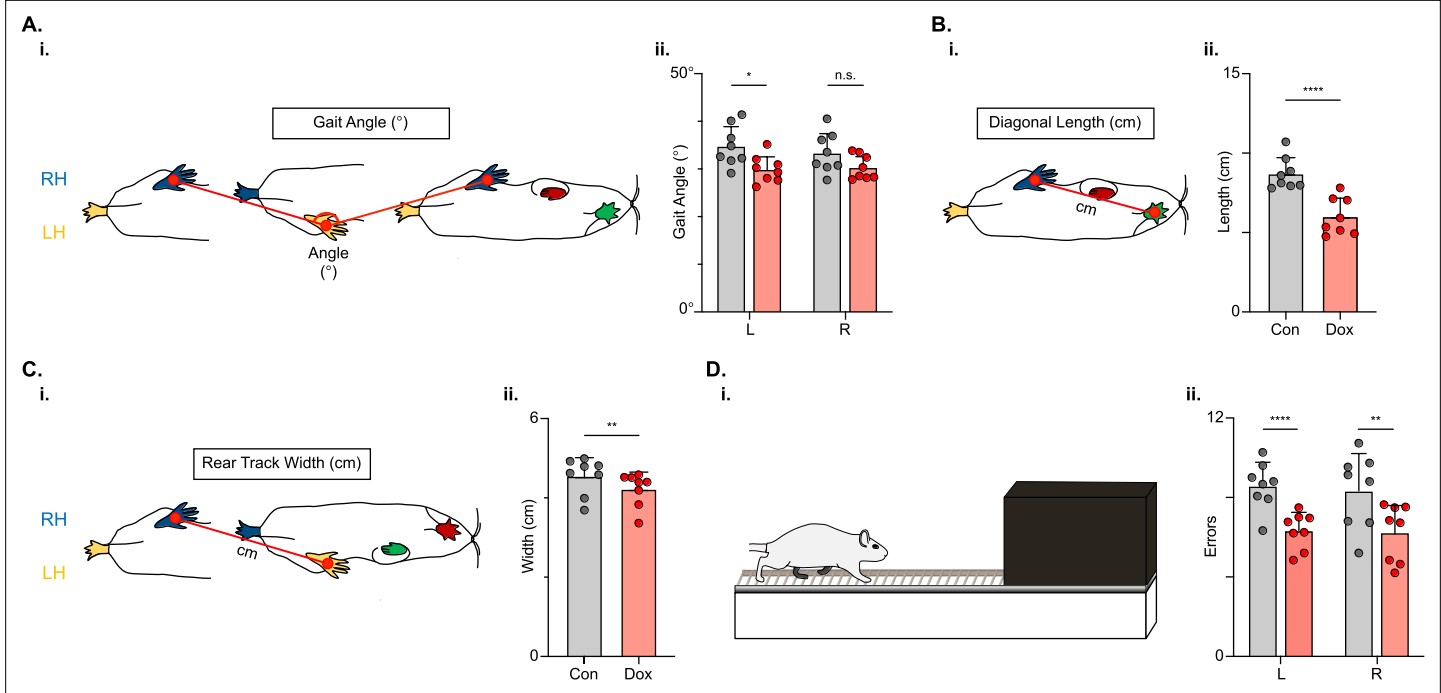

**Figure 10.** Indicators of postural stability were mildly improved during post-spinal cord injury (SCI) long descending propriospinal neuron (LDPN) silencing. Throughout this figure, the right hindlimb (RH) will be shown in blue, the left hindlimb (LH) will be shown in yellow, the right forelimb (RF) will be shown in red, and the left forelimb (LF) will be shown in green. Control values are indicated by gray bars and Dox values are indicated by pink bars. Individual animal averages are shown for each postural stability variable are indicated by gray circles (Control) and red circles (doxycycline, Dox). An example of gait angle from a single set of initial paw contacts for the right hindlimb in reference to the left hindlimb is demonstrated in **A**ᵢ. The average gait angle was calculated for both the left and right hindlimbs separately (**A**ᵢᵢ; Control left 34.81±4.26 vs Dox left 29.95±2.82, df = 7, *p<0.05; Control right 33.37±4.21 vs Dox right 30.35±2.54, df = 7, n.s.; RM ANOVA). An example of diagonal length, calculated from the initial contact of each hindlimb to its contralateral forelimb, is also shown between the right hindlimb and the left forelimb (**B**ᵢ). Averages for Control and Dox are demonstrated in (**B**ᵢᵢ; Control diagonal length: 8.70±1.02 vs Dox diagonal length: 6.00±1.16, t=6.96, df = 7, ****p<0.001; paired t-test). Rear track width, calculated as the distance between each hindlimb's initial contact, is exemplified in (**C**ᵢ), with averages seen in (**C**ᵢᵢ; Control rear track width: 4.55±0.47 vs Dox rear track width: 4.22±0.42, t=3.99, df = 7, **p=0.005; paired t-test). Finally, the fixed-rung setup for ladder testing is shown in (**D**ᵢ). The black box indicates a dark box found at the end of the ladder, which is used as an incentive for rats to traverse the apparatus. The average footfalls were calculated for both the left and right hindlimbs separately (**D**ᵢᵢ; Control left 8.58±1.19 vs Dox left 6.34±0.92, df = 7, ****p<0.001; Control right 8.33±1.88 vs Dox right 6.24±1.43, df = 7, **p<0.01; RM ANOVA).

The online version of this article includes the following source data for figure 10:

**Source data 1.** File contains the raw data for *Figure 10* showing the paw placement position and calculated gait angle, rear track width, and diagonal step length for post-injury animals with and without doxycycline (Dox).

---

hallmark of LDPN silencing in uninjured animals, or changes in hindlimb-forelimb coordination. Collectively, these findings suggest that rather than securing alternation of the hindlimbs and forelimbs, spared LDPNs hinder the capability of lumbar circuitry below the level of the lesion to function optimally, contributing to diminished stepping capacity at chronic post-injury time points. No doubt, these results are counter-intuitive because any decrease in spared axons that cross the site of injury should have a negative impact on locomotor function/recovery.

However, these results follow two recent studies from our laboratory focused on LAPNs, the ascending equivalent to the LDPNs (*Pocratsky et al., 2020*; *Shepard et al., 2021*). LAPN silencing in uninjured animals disrupted interlimb coordination that mimicked the findings with LDPN silencing, but with several important differences. Perturbations with LAPN silencing were more robust, context-specific, and had a more even disruption to both forelimb and hindlimb pairs. Neither manipulation significantly influenced interlimb coordination during swimming, intralimb coordination, or speed-dependent gait characteristics during stepping within the limitations of the assessment strategy. Taken together with our anatomical analysis of these two populations (*Pocratsky et al., 2020*; *Shepard et al., 2021*), our results lead us to suggest that LAPNs and LDPNs form an inter-enlargement loop,

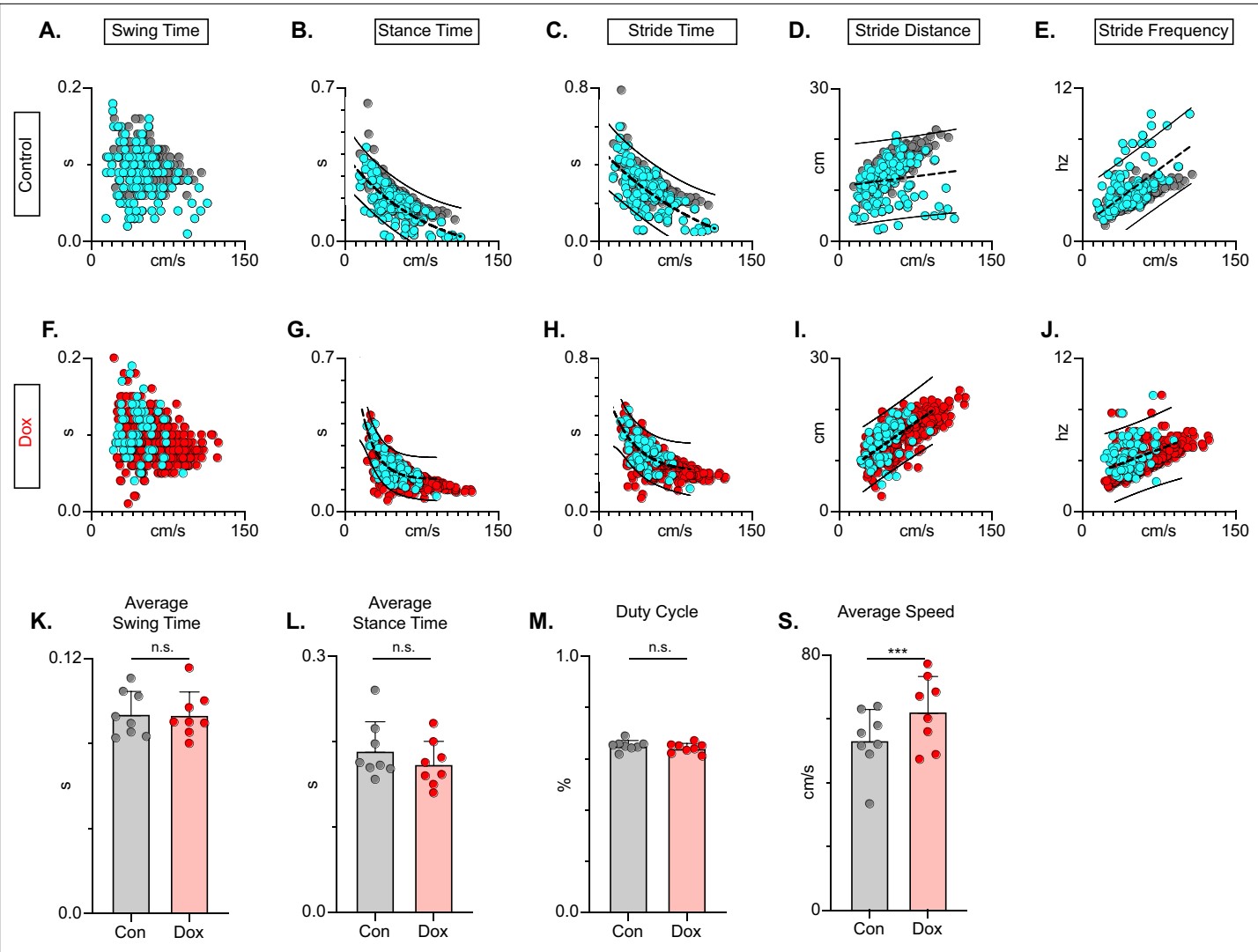

**Figure 11.** Long descending propriospinal neuron (LDPN) silencing restores key features of locomotion are restored following post-SCI. Relationships between swing time, stance time, stride time, and stride distance are plotted against speed for Control (**A-E**) and doxycycline (Dox) (**F-J**) time points. Dorsal steps are indicated with teal circles for both Control and Dox, while plantar steps are indicated with either gray or red circles for Control and Dox, respectively. An exponential decay trendline is displayed for stance time and stride time graphs, while a linear trendline is displayed for stride frequency and stride distance (dotted line indicates trendline; Stance time: Control $R^2$=0.476 vs Dox $R^2$=0.638; Stride time: Control $R^2$=0.406 vs Dox $R^2$=0.556; Stride distance: Control $R^2$=0.194 vs Dox $R^2$=0.458; Stride Frequency: Control $R^2$=0.259 vs Dox $R^2$=0.495). 95% prediction intervals are shown for lines of best fit as solid lines. Average swing time (**K**, Control swing time 0.094±0.011 vs Dox swing time: 0.093±0.010, t=0.17, df = 7, n.s., paired t-test) and average stance time (**L**, Control stance time 0.189±0.034 vs Dox stance time: 0.174±0.027, t=1.94, df = 7, n.s., paired t-test) are indicated with circles representing individual animal averages. The average duty cycle (stance time/stride time **M**, Control duty cycle 0.651±0.020 vs Dox duty cycle: 0.641±0.019, t=1.83, df = 7, n.s., paired t-test) and average speed (**S**, Control speed 53.33±9.67 vs Dox speed: 62.31±11.05, t=4.177, df = 7, ***p<0.005, paired t-test) are plotted for Control (gray) and Dox (red) time points with averages indicated by bars. 1 SD is indicated by the bar above the average.

The online version of this article includes the following source data for figure 11:

**Source data 1.** File contains the raw data for *Figure 11* showing the spatiotemporal components of stepping including swing, stance, and stride times vs speed, stride distance and frequency vs speed, plus the average swing and stance times, the duty cycle, and average speed.

providing temporal information used to secure R-L coordination within each enlargement. Clearly, the context specificity of the LAPN-silenced phenotype argues that sensory input is critical, perhaps to trigger enhanced supraspinal oversight when stepping conditions are not optimal. The speculation that these two populations are part of an anatomical and functional inter-enlargement network is further supported by the similar impact of post-SCI synaptic silencing. When presumably spared LAPNs or LDPNs were silenced post-SCI, we observed significant improvements in stepping. In the

case of the LAPNs, the precision of R-L hindlimb alternation was essentially restored to pre-injury levels, and the incidence of dorsal stepping was greatly reduced. The impact of LDPN silencing post-SCI was less pronounced, but still resulted in measurable improvements in both interlimb coordination and trunk stability. Taken together, these results suggest that disruption of an already incomplete (due to the loss of white matter at the injury epicenter) ascending-descending circuit frees-up or permits the lumbar spinal circuitry to function more autonomously, unimpeded by incomplete or noisy information.

Interestingly, the thoracic contusion injury utilized here and previously (NYU 12.5 g-cm; *Shepard et al., 2021*) resulted in a disruption to hindlimb alternation similar to that seen during LDPN silencing but did not disrupt forelimb alternation. Silencing selectively removes the targeted LAPNs or LDPNs, whereas a contusive injury is anatomically selective for the dorsal columns (including the CST), lateral, ventrolateral, and ventral white matter with a medial to lateral bias. This then suggests that the loss of the dorsal columns plus some of the LAPN or LDPN axons (following injury) results in an increased supraspinal oversight of forelimb coordination that ignores or is resilient to the additional impact of LAPN or LDPN silencing. Forelimb alternation is intact at all time points post-thoracic contusion, whereas hindlimb alternation is disrupted by the thoracic contusion or by LAPN or LDPN silencing.

*Ruder et al., 2016* reported that in the adult mouse, ablation of LDPNs using intersectional approaches reduced the speed and duration of spontaneous locomotor bouts, and disrupted interlimb coordination during high-speed locomotion on a treadmill. In contrast, our data indicate that, despite the disrupted alternation of the two limb pairs, rats are still able to achieve fairly high rates of speed and that locomotor bouts are sustained and unperturbed when LDPNs are silenced in otherwise intact adults. These discrepancies may arise from several differences between the two studies. First, the specific neurons targeted by Ruder et al. may have included LDPNs from multiple cervical segments whereas we chose to focus specifically on LDPNs with cell bodies at C6. Second, the ablation of neurons using the genetic approach (with diphtheria toxin) is permanent and may lead to unrecognized circuit reorganization in the adult.

The LDPN population targeted in the current study may be comprised of different sub-populations of long spinal projection neurons that have been previously defined genetically in mice. Notably, V0v and V2a, but not V0d, neurons residing in the lumbar enlargement have been implicated in securing left-right limb alternation at high speeds (*Bellardita and Kiehn, 2015*; *Crone et al., 2009*; *Talpalar et al., 2013*; *Ruder et al., 2016*). V0v are composed of commissurally-projecting inhibitory interneurons and are recruited to maintain left-right alternation as the speed of movement increases (*Lanuza et al., 2004*; *Crone et al., 2009*). The V2a population of neurons are also classified as ipsilateral excitatory glutamatergic interneurons that are involved in left-right alternation (*Crone et al., 2008*; *Crone et al., 2009*) with minimal involvement in rhythm generation. It is likely that the silenced LDPN population includes some proportion of each of these defined interneurons, given the perturbations to left-right alternation during silencing. However, it is important to note that LDPNs have characteristics that fall in line with both progenitor domains, as they are both ipsilaterally and commissurally-projecting and co-localize with both glutamatergic and GABAergic synapses. Therefore, it is difficult to place these anatomically-defined neurons within the previously described context of genetically-defined V-neuron classes.

By virtue of their role in communicating higher motor commands to spinal circuits, a major focus of spinal cord repair has been the regeneration of descending tracts across the lesion site to restore lost motor input (*Tuszynski and Steward, 2012*). As a consequence of the LDPN cell body position within the cervical spinal cord and their central role in the generation of patterned locomotor output prior to the injury, LDPNs are well-suited to receive synaptic contacts from descending supraspinal projections to form 'detour circuits' that might propagate (i.e. relay) supraspinal motor system commands to below the level of injury (*Bareyre et al., 2004*; *Courtine et al., 2008*; *Mitchell et al., 2016*). Consistent with that suggestion, evidence continues to point to these descending propriospinal neurons as potential targets for functional recovery (*Taccola et al., 2018*; *Loy and Bareyre, 2019*). Our targeted population of LDPNs represents a partially-spared inter-enlargement pathway that should continue to reliably carry information across the level of injury.

Another important consideration is the different types of circuitry that provide input to this propriospinal neuronal population. Early electrophysiological data showed that supraspinal motor centers in the cerebellum, cerebral cortex, and brainstem, along with primary afferents, provide monosynaptic

input onto LDPN populations in the cervical spinal cord (*Brink et al., 1985*; *Alstermark et al., 1987*). More recent work has confirmed the presence of corticospinal and descending serotonergic pathways in LDPNs (*Ni et al., 2014*). *Flynn et al., 2017* showed that commissurally-projecting LDPNs received putative synaptic inputs from both excitatory and inhibitory pathways, such as inhibitory pre-motor interneurons, excitatory corticospinal or myelinated afferents, and proprioceptive input from group Ia muscle afferents. Our data showing IHC colocalization with GABA and glutamate (*Figure 6*) are consistent with those data, but the specific pathways with which inter-enlargement LDPNs interact is not apparent and requires further investigation.

At first glance, our results contradict several studies in the field of locomotor recovery in which various populations of descending propriospinal relays of both thoracic and cervical origin are seen as an essential component of recovered stepping ability (*Bareyre et al., 2004*; *Vavrek et al., 2006*; *Flynn et al., 2011*; *Courtine et al., 2008*; *Courtine et al., 2008*). However, the descending populations in these studies are inconsistent in terms of their defined anatomy (location of cell bodies and terminals, *Figure 1a*). This is a critical consideration, as even minor differences in the anatomical location of spared pathways may drastically alter their influence on below-level circuitry and their propensity to enhance recovery after SCI. The extensive differences in how LDPNs are defined may also offer some clues to the described differences in recovered function described here. Descending pathways with more rostral cell bodies within the cervical cord (*Bareyre et al., 2004*) may have an entirely different function than those located more caudally. Others refer to LDPNs without specifying defining anatomical characteristics (*Vavrek et al., 2006*; *Courtine et al., 2008*; *Filli et al., 2014*). The ambiguity surrounding long propriospinal neurons complicates the direct comparison of results regardless of how the neuronal activity is being modulated.

Some of the referenced studies have also focused specifically on the potential of supraspinal axon regeneration/sprouting and synapse formation onto LDPNs (*Vavrek et al., 2006*; *Courtine et al., 2008*). Whether this regeneration/sprouting leads directly to LDPN-dependent improvements in functional outcomes remains uncertain. Our results suggest that spared LDPNs are detrimental to recovered locomotion and thus may not be good targets for regenerative/bridging efforts. Definitive identification of descending propriospinal pathways that do facilitate functional recovery after SCI is essential to understand therapeutically functional plasticity.

## Materials and methods

Experiments were performed in accordance with the Public Health Service Policy on Humane Care and Use of Laboratory Animals, and with the approval of the Institutional Animal Care and Use (IACUC) and the Institutional Biosafety (IBC) Committees at the University of Louisville.

### Experimental design

Animals were housed two per cage under a 12 hr light/dark cycle with ad libitum food and water throughout the course of the study. Power analysis for gait measures revealed that n=6–10 could detect a true significant difference with a power of 85–95% and an alpha of 0.05. Additional animals were used to mitigate animal mortality following repeat surgical exposure. A total of n=19 adult female Sprague-Dawley rats (215–230 g) were used in this study (*Figure 1c*). After viral vector injection and initial Dox administration ('Dox1'; n=19), animals were split into two groups. Animals were assigned into the uninjured (n=8) and injured groups (n=11) with their cage mate after Dox1 administration and behavioral assessments. The uninjured group received one additional round of Dox (Dox2) lasting 13 days and were subsequently euthanized. In the injured group, n=1 animals died as a result of SCI injury before post-injury behavioral assessments. Eight of 10 remaining animals displayed a locomotor phenotype pre-SCI. The two animals that did not display behavioral changes were removed from the main dataset to be analyzed separately as a post-injury control. These animals had fewer than 10% of their steps outside of normal variability during Dox testing. Animals that lacked behavioral outcomes pre-injury (n=2) did not show improvements in locomotion post-SCI. However, both animals had mild control injury phenotypes, clouding deeper interpretations of post-injury silencing data for these animals.

### Viral vector production

Viruses were constructed and titered following previously described methods (*Pocratsky et al., 2017*; *Pocratsky et al., 2020*).

### Intraspinal injections of viral vectors to doubly infect LDPNs

Intraspinal injections and power analyses for kinematic measures are based on previous literature. The procedural details are described in the *Nature Protocol Exchange* (*Pocratsky et al., 2018*).

### Spinal cord injury

Animals received a moderate spinal cord contusion at T9/T10 spinal cord approximately 2 weeks after the conclusion of uninjured Dox[ON] assessments. For SCI surgery, animals were re-anesthetized (ketamine:xylazine:acepromazine, 40 mg/kg:2.5 mg/kg:1 mg/kg; I.P., Henry Schein Animal Health, Dublin, OH; Akorn Animal Health, Lake Forest, IL). Spinal cord contusion injuries were performed as described in our previous paper (*Shepard et al., 2021*).

### Experimental timeline

Doxycycline hydrochloride (Dox, 20 mg/ml; Fisher Scientific BP2653-5, Pittsburgh, NH) was dissolved in 3% sucrose and provided ad libitum for 8 days pre-injury and 8–14 days post-injury. Dox water was made fresh and replenished daily and monitored for consumption. All behavioral assessments were performed during the light cycle portion of the day and concluded several hours before the dark cycle began.

Prior to SCI, behavioral assessments were performed prior to viral injections (BL), prior to Dox (PD1), during Dox (D1D5-D1D8), and 14 days post-Dox (PostD1). For the uninjured LDPN animals, behavioral assessments were performed again prior to the second Dox administration (PD2) and during Dox (D2D5, D2D8, D2D13). Following SCI in the injured group, pre-Dox and Dox[ON] time point assessments were reproduced twice (Dox2 and Dox3) following SCI to assess the reproducibility of any behavioral changes that were seen. Post-injury control time points included assessments prior to Dox (PD2, PD3), during Dox (D2D5, D2D8, D3D5, D3D8, and D3D14), and after the second Dox administration (PostD2). Some data shown are compiled from pre-injury Control and Dox time points and post-injury Control and Dox time points. Control vs Dox uninjured and Control vs Dox injured time point comparisons were made both on an individual and group basis. Behavioral analyses began on day 5 of Dox (DoxD5) administration and were repeated on Dox day 8 (DoxD8). For terminal assessments after injury, behavioral assessments occurred on Dox Day 14 (D3D14).

### Identification of hindlimb joints, hindlimb kinematics, and intralimb coordination analysis

Hindlimb joint identification and hindlimb kinematics were acquired as previously described (*Pocratsky et al., 2020*; *Shepard et al., 2021*). The analyzed points from the two sagittal cameras were exported to a Microsoft Excel workbook and 2D average angles were calculated for each digitized frame. Maximum and minimum angles (maximum extension and flexion, respectively) were identified using a custom Microsoft Excel macro. Excursion for the proximal and distal hindlimb joint angles (Maximum Angle – Minimum Angle) were also calculated using the same macro. The temporal relationship between the proximal and distal hindlimb joint angles was calculated using the peak-to-peak duration of the lead angle during a single-step cycle. Within this duration, the maximum excursion of the first angle was determined and depended on which angle peaked first (proximal or distal angles). The time of onset of the second angle was divided by the peak-to-peak distal angle duration to determine the temporal relationship between intralimb angles. A coordination value of 1 indicates in-phase coordination of the proximal and distal intralimb joints, while a phase value of 0.5 indicates anti-phase, uncoordinated joint movements. Intralimb joint phase was calculated for each step cycle of the left and right hindlimbs independently.

### Overground gait analyses

Overground gait analyses were performed as described previously (*Shepard et al., 2021*). During recordings of injured animals (control and silenced), a step was classified as dorsal when the dorsum of the foot came into contact with the ground during the stance portion of the step cycle and is considered a step if it maintains contact with the surface and completes the swing portion of the step cycle. For both uninjured and injured conditions, gait analysis was always performed relative to the left hindlimb (i.e. RLRR and RLFR). Consequently, for the overground gait (phase) analysis dorsal steps were identified only for the left hindlimb and any dorsal steps on the right side would not be identified

in the analyzed dataset. In separate analysis (CPI, DSI, right-left dorsal step ratio) dorsal steps for both right and left hindpaws were considered (see *Figure 7*).

Interlimb phase was calculated by dividing the initial contact time of the trailing limb by the stride time (initial contact to initial contact) of the leading limb. Alternating gaits were defined by phase values concentrated around 0.5 (*Lemieux et al., 2016*).

Phases for individual steps were transformed from a circular scale and were plotted linearly. Blue boxes on graphs represent >2 standard deviations (SD) as calculated from the uninjured control average and SD for the specified limb pair. Values found within the blue box are considered outside of normal variability and were quantified using pie charts to indicate the percent of total steps that existed outside of that range. Plantar steps are indicated by gray circles during control and red circles during silenced time points. Post-injury dorsal steps were indicated on both circular and linear phase plots as teal circles.

## BBB assessments

BBB assessments were performed by individuals blinded to experimental time points. All raters were aware that assessments were being performed post-injury, but were blind to Control and Dox. Overground stepping was assessed using the BBB Open Field Locomotor Scale as previously described (*Basso et al., 2002*; *Caudle et al., 2015*).

## Coordination indices

RI, CPI, PSI, and DSI were calculated as described previously (*Shepard et al., 2021*).

## Postural stability

Several measures were used to determine the postural stability of animals during locomotor bouts. Gait angle is the angle between two consecutive initial contacts of a rear hindlimb in reference to the other hindlimb. Typically, a smaller gait angle would indicate higher stability during locomotion, as the hindlimbs are in the line with the body. Larger gait angles would suggest diminished stability as the hindlimbs would be set wider. Gait angle is calculated using the Pythagorean theorem and the X, Y coordinates of the foot while it is in contact with the ground. Statistics were performed on the group means for the left and right hindlimbs (bars: average ± SD; circles: individual means).

The diagonal length is the distance between the initial contacts of diagonal paws. For this calculation, the diagonal pair used was left hindlimb–right forelimb. This pair was chosen to remain consistent with hindlimb phase data in which the left hindlimb is the lead limb. The diagonal length is calculated using the Pythagorean Theorem and the X, Y coordinates of each foot.

Rear track width is the distance between the hindlimbs during consecutive initial contacts. Track width is calculated by taking the absolute value of the Y-coordinate of the left hindlimb initial contact subtracted from the Y-coordinate of the subsequent right hindlimb initial contact. Larger track width indicates a wider stance during stepping, while a smaller track width indicates stance phases more in line with the body.

## Ladder

We quantified the animals' ability to effectively traverse a ladder with fixed-spacing rungs (Columbus Instruments, Columbus OH, *Chen et al., 2012*; *Metz and Whishaw, 2002*). Behavioral testing was performed on the same time points as BBB testing (PD2, D2D5, D2D8, PostD2, PD3, D3D5, D3D8, and D3D13). Each animal received five stepping trials in each direction. The total number of footfalls was calculated for the left and right hindlimbs, respectively, for each animal across the time points. We then calculated each animal's average number of foot slips during the Control and Dox time points listed above. Left and right limbs were not combined to demonstrate that the number of foot slips decrease by a similar amount for each hindlimb (i.e. one foot is not different from the other in terms of recovery). Statistics were performed on the group means (bars: average ± SD; circles: individual means).

## Spatiotemporal gait indices

Spatiotemporal gait analyses were performed as described previously (*Shepard et al., 2021*).

## Histological analyses

Animals were killed on D3D14 (n=10) following terminal BBB and kinematic behavioral assessments. Animals were overdosed with sodium pentobarbital, followed by pneumothorax and transcardial perfusion with 0.1 M phosphate-buffered saline (PBS) (pH 7.4) followed by 4% paraformaldehyde diluted in PBS solution. Spinal cords were dissected, post-fixed for 1.5 hr, and transferred to 30% sucrose for a minimum of 4 days at 4 °C. Spinal segments C5-C8, T8-T12, and T13-L3/L4 were dissected, embedded in tissue freezing medium, and stored at –20 °C until they were cryosectioned at 30 μm.

Histological analysis was performed using the immunohistochemistry protocol described previously (*Shepard et al., 2021*).

## Statistical analyses

Statistical analyses were performed using the SPSS v22 software package from IBM. Additional references for parametric and non-parametric testing were used in complementation to SPSS (*Hays, 1981*; *Siegel and Castellan, 1988*; *Ott and Longnecker, 1977*). Differences between groups were deemed statistically significant at p≤0.05. Two-tail p-values are reported.

The Binomial Proportion Test was used to detect significant differences in the proportion of coordination values beyond the control threshold for the raw and transformed interlimb coordination data of various limb pairs prior to and post-SCI. It was also used to determine statistical significance for per-step changes in left-right coordination and change in interlimb phase, raw BBB score differences, intralimb phase and per-step changes in intralimb phase, dorsal steps as a percentage of total steps, and percentage of categorically organized steps (anti-phase, out of phase, in phase).

Regression analyses were used to compare speed versus spatiotemporal gait indices datasets, including speed vs. swing time, stance time, stride time, and stride distance. Analyses were performed for hindlimb-hindlimb relationships prior to and after SCI. For regression analyses post-SCI, plantar, and dorsal steps were included in the analysis and dorsal steps are shown in blue on graphs for identification. Trend lines are shown on graphs with 95% prediction intervals indicated by dashed lines.

Mixed model analysis of variance (ANOVA) followed by Bonferroni *post hoc* t-tests (where appropriate) were used to detect a significant difference in the BBB scores based on sidedness and condition (i.e. Control vs Dox; graph not shown). Repeated measures ANOVA analyses were used to compare the number of average steps per animal during uninjured and injured time points.

Paired t-tests were used to detect significant differences in proximal and distal angle excursion for uninjured and injured intralimb coordination, average gross stepping measures including RI, CPI, PSI, DSI, and combined BBB scores, average intralimb phase, percentage of dorsal step sidedness, average swing time, average stance time, average duty cycle, average number of ladder errors, average gait angle, average diagonal length, average rear track width, and overall average speed at Control and Dox combined time points.

Circular statistics were performed on the raw phase data to analyze phase distribution of forelimb, hindlimb, heterolateral and homolateral limb pairs as described previously (*Pocratsky et al., 2017*). Time point comparisons were performed using the nonparametric two-sample U2 tests. The p-values are reported as ranges based on critical values of Watson's U2 as reported in Table D.44 of *Zar, 1974*.

## Acknowledgements

HHS | NIH | National Institute of Neurological Disorders and Stroke (NINDS): Scott R Whittemore, David SK Magnuson, R01 NS112304-01; HHS | NIH | National Institute of Neurological Disorders and Stroke (NINDS): Scott R Whittemore, David SK Magnuson, R01 NS089324; HHS | NIH | National Institute of Neurological Disorders and Stroke (NINDS): Brandon L Brown, F31 NS116935. The funders had no role in study design, data collection and interpretation, or the decision to submit the work for publication.

## Additional information

### Funding

| Funder | Grant reference number | Author |
| --- | --- | --- |
| National Institute of Neurological Disorders and Stroke | R01 NS112304-01 | David SK Magnuson<br>Scott R Whittemore |
| National Institute of Neurological Disorders and Stroke | R01 NS089324 | David SK Magnuson<br>Scott R Whittemore |
| National Institute of Neurological Disorders and Stroke | F31 NS116935 | Brandon L Brown |

The funders had no role in study design, data collection and interpretation, or the decision to submit the work for publication.

### Author contributions

Courtney T Shepard, Conceptualization, Data curation, Formal analysis, Validation, Investigation, Visualization, Methodology, Writing – original draft; Brandon L Brown, Data curation, Visualization, Methodology, Writing – review and editing; Morgan A Van Rijswijck, Rachel M Zalla, Johnny R Morehouse, Data curation, Formal analysis, Methodology; Darlene A Burke, Data curation, Formal analysis; Amberly S Riegler, Data curation, Formal analysis, Visualization, Project administration; Scott R Whittemore, Conceptualization, Supervision, Funding acquisition, Writing – review and editing; David SK Magnuson, Conceptualization, Supervision, Funding acquisition, Project administration, Writing – review and editing

### Author ORCIDs

Scott R Whittemore ⓘ http://orcid.org/0000-0001-6437-7200
David SK Magnuson ⓘ https://orcid.org/0000-0003-3816-3676

### Ethics

This study was performed in strict accordance with the recommendations in the Guide for the Care and Use of Laboratory Animals of the National Institutes of Health. All of the animals were handled according to approved institutional animal care and use committee (IACUC) protocols (#19644) of the University of Louisville. All surgery was performed under ketamine/xylazine anesthesia supplemented with isoflurane, and every effort was made to minimize suffering.

### Decision letter and Author response

Decision letter https://doi.org/10.7554/eLife.82944.sa1
Author response https://doi.org/10.7554/eLife.82944.sa2

## Additional files

### Supplementary files

• Transparent reporting form

### Data availability

All source data for the main figures have been provided.

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
