## [Editor Report]

This paper evaluates the roles of neurons arising in the spinal cord in the cervical (neck) regions that extend axons to lumbar regions that control the legs and facilitate recovery of walking ability after spinal cord injury. The paper is important because it provides evidence that neurons arising in the neck do not help the recovery of hindlimb function and in fact, mildly impair it. Most of the evidence is convincing although some limitations were noted. The data adds new information on the role of long projecting interneurons in the spinal cord affecting limb coordination during locomotion and how their silencing helps restore partial function after spinal cord injury.

---

## [Decision Letter]

**Decision letter after peer review:**

Thank you for submitting your article "Silencing long-descending inter-enlargement propriospinal neurons improves hindlimb stepping after contusive spinal cord injuries" for consideration by *eLife*. Your article has been reviewed by 3 peer reviewers, including Christopher Cardozo as Reviewing Editor and Reviewer #1, and the evaluation has been overseen by Mone Zaidi as the Senior Editor.

Essential revisions:

When considering how to revise the manuscript please pay particular attention to the following points.

1. Add one to two sentences that expand on how the viral system used for tet-inducible synaptic silencing works and is used.

2. Consider further whether it is possible to quantify the efficiency of viral transduction and how variability in efficiency might influence the interpretation of results.

*Reviewer #1 (Recommendations for the authors):*

Clarification of a few points would help the reader:

Line 215: why was this location chosen for histology if the injections were done at C2?

Line 229: avoid the double negative.

Line 400 or so: could an alternative be that changes in the excitability of local circuits change how information from LDPN is processed? Just a consideration you may wish to discuss.

Line 497: Please indicate the α used for these calculations.

Line 498: Please consider including the number added to replace lost animals.

Line 566: Consider rephrasing this. Does this focus the presentation without detracting from completeness since no difference was seen between L and R? In these days of omics huge datasets have become the norm.

Please review all figure legends to be sure that the meaning of the asterisks shown in the figures is clearly defined.

*Reviewer #2 (Recommendations for the authors):*

It was not evident why there was not more of a focus on homolateral and heterolateral forelimb/hindlimb coupling. When one thinks about LDPNs, the obvious changes would be expected to be in the forelimb-hindlimb coupling, rather than left-right at lumbar and cervical levels. A disruption in either homolateral or heterolateral coupling is also likely to disrupt left-right coupling. It is understandable that left/right coupling is featured prominently due to the results from the LAPNs but forelimb-hindlimb coupling should also be displayed. Based on Figure 2 supp 1, it looks like heterolateral coupling is disrupted by silencing but not homolateral.

The single-figure examples of terminals in Figure 6 are not entirely convincing. Some sort of quantification of terminals should be possible in both SCI and uninjured cords. It is also difficult to interpret that data without knowing the approximate percentage of LDPNs that were silenced (or that expressed TeTN).

*Reviewer #3 (Recommendations for the authors):*

This study explores the role of descending propriospinal neurons on inter-limb coordination during locomotion and how their silencing helps improve some parameters after spinal cord injury. Below are specific concerns and comments on this study.

Results

1) Line 96: please clarify how the double infection was performed, which viruses were injected and how did you ensure that neurons were silenced. Figure 1 only outlines the experimental procedure and does not provide any data in this sense. How do the authors control that TeNT is reversible and how long does it take to reverse the silencing?

2) Were the injected viruses tagged with fluorescent reporters? It would be helpful to provide some quantification on the number of DLPNs neurons and their distribution in the spinal cord. Are they restricted to C6?

3) Figure 2: please define the abbreviations used in the graphs. Why were values from each trial in the same animal presented rather than the mean value per animal? Is this statistically correct?

4) It is not clear to this reviewer why and how silencing of DLPNs in lumbar segments affects the coordination of forelimbs.

4) Figure 3, graphs A-J: it is more relevant to plot data points from control Acryl and Dox Acryl together and control Sylgard and Dox Sylgard together in the same graphs. This will allow the reader the appreciate potential differences. Here again, it is not clear why data points from each trial were presented rather than mean values per animal.

5) Figure 5: the figure and text need more clarification. The authors should indicate in the different panels the parts of the white matter that have been spared and the location of the descending projections. It is not clear how the authors come to the conclusion that the LDPN axons have been spared while only 26% of white matter was spared.

6) Figure 6: The authors state that GFP is expressed in cell bodies of neurons in the cervical spinal cord. However, the staining looks more like terminals contacting the stained cell bodies rather than the expression of GFP. Quantifications are necessary here to support the authors' conclusions.

Discussion

4) In the discussion, the authors state that the study by Ruder et al. (2016) used developmental ablation of subsets of LDPNs. Ruder et al. used viral infections to express DTR in descending propriospinal neurons in adult mice. The difference in the outcome of this study compared to that of Ruder et al. cannot be explained by circuit reorganization during development. In the present study, there is no information on the extent of silencing and which neuronal populations are affected. This part of the discussion does not reflect the existing data and does not explain the difference in the outcome between this study and Ruder et al. (2016).

5) Overall, it is not clear how silencing of DLPNs restores some parameters after spinal cord injury and how relevant this approach would be clinically.

[Editors’ note: further revisions were suggested prior to acceptance, as described below.]

Thank you for resubmitting your work entitled "Silencing long-descending inter-enlargement propriospinal neurons improves hindlimb stepping after contusive spinal cord injuries" for further consideration by *eLife*. Your revised article has been evaluated by Timothy Behrens (Senior Editor) and a Reviewing Editor.

The manuscript has been improved but there are some remaining issues that need to be addressed, as outlined below:

1. Please carefully check and revise the text discussing how your findings relate and compare to those reported by Ruder (2016). In particular, please note that in the Ruder paper, some experiments looking at neural circuits were conducted based upon tamoxifen-induced Cre-mediated recombination at E10.5 (Figure 1) but it appears to us that experiments examining functional outcomes were after manipulating gene expression by injecting viral constructs after birth. Thus, when discussing how ablating subsets of LDPNs please be sure you are referencing and discussing the correct experimental system.

2. Thank you for informing us of the errors in some of the figures. Please include updated figures with this resubmission and, update figure legends if needed.

---

## [Author Response]

Essential revisions:When considering how to revise the manuscript please pay particular attention to the following points.1. Add one to two sentences that expand on how the viral system used for tet-inducible synaptic silencing works and is used.

In response to this concern the following sentences have been added to the beginning of the Results section. It is highlighted in the manuscript.

“For this study we employed a two-virus synaptic silencing strategy developed by Tadashi Isa and colleagues (Kinoshita et al., 2012). This system involves a highly efficient lentiviral vector (HiRet) with a tetracycline response element upstream of enhanced tetanus neurotoxin (eTeNt) and EGFP (eTeNT.EGFP) that infects neuron terminals. Next, an AAV2/2 virus that contains a tetracycline transactivator (rtTAV16) is delivered which infects neuron cell bodies. Doxycycline (DOX) then induces the production of eTeNT/EGFP which cleaves the vesicular docking protein VAMP2 thus preventing neurotransmitter release and silencing the neurons based only on their anatomy, any/only neurons with terminals exposed to the lentiviral vector and cell bodies exposed to the AAV2/2 will be silenced, as employed previously (Pocratsky et al., 2017, 2020; Shepard et al., 2021). To silence the LDPNs”

2. Consider further whether it is possible to quantify the efficiency of viral transduction and how variability in efficiency might influence the interpretation of results.

Given that this is a research advance with 2 other previous *eLife* publications using the identical synaptic silencing system, I will include a slightly modified version of the response to reviews we provided previously:

“A recent publication from our laboratories (PMID: 33828464) found similar numbers of ipsilateral LAPNs were labeled when using Fluororuby, a 10kD rhodamine dextran amine, (mean number of LAPNs labeled = 135 +/-52) and a dual-viral system (mean number of LAPNs labeled = 126 +/- 46) similar to the dual-viral silencing a silencing system used here. Provided Fluororuby gives an accurate representation of the number of LAPNs at/near the injection sites, the dual-viral silencing system used here likely silences >90% of LAPNs (defined as neurons with cell bodies at L2 and axon terminals at C6). Additionally, eTeNT (produced by the dual-viral silencing system) has extremely high catalytic activity (PMID: 7527117) that is active at very low levels. Which also supports that >90% of LDPNs were silenced in the current study.”

We agree that the variability in efficiency of silencing could influence the results quantitatively (the robustness of the expressed phenotype) but believe it would be less likely to influence the results qualitatively (some other primary disruption to stepping). In other words, the partial decoupling of the fore and hindlimb pairs in the intact animal and the modest improvement in interlimb coordination when silenced post-injury would very likely remain the same, just more or less robust that what we have observed. Thus, our overall interpretation of the results would be unlikely to change.

Reviewer #1 (Recommendations for the authors):Clarification of a few points would help the reader:Line 215: why was this location chosen for histology if the injections were done at C2?

Injections were at L2 and C6/7, and here we refer to caudal cervical segments generally, meaning C6/7.

Line 229: avoid the double negative.

Thanks for this suggestion, it has been fixed.

Line 400 or so: could an alternative be that changes in the excitability of local circuits change how information from LDPN is processed? Just a consideration you may wish to discuss.

We have certainly considered this possibility and chose not to bring it up in this paper or in the previous LAPN paper because post-DOX the behavior always returns to the pre-DOX levels. In our thinking, if the silencing were to induce changes in excitability of the circuitry below the injury, then these changes should manifest as a difference between pre- and post-DOX function/behavior. I hope this makes sense.

Line 497: Please indicate the α used for these calculations.

The α of.05 has been added.

Line 498: Please consider including the number added to replace lost animals.

We apologize for the confusing sentence (lines 497-8). We have edited that section as follows:

“In order to mitigate the impact of animal mortality associated with repeated surgeries, a total of N=19 adult female Sprague-Dawley rats (215-230 g) were entered into this study (Figure 1C).”

This is shown highlighted in the manuscript.

Line 566: Consider rephrasing this. Does this focus the presentation without detracting from completeness since no difference was seen between L and R? In these days of omics huge datasets have become the norm.

Thank you for catching this confusing section. We have edited this section to clearly indicate that the left-side only dorsal stepping was only for the phase analysis, but that both sides were considered for the other analysis. The section now reads:

“For both uninjured and injured conditions gait analysis was always performed relative to the left hindlimb (ie. RLRR and RLFR). Consequently, for the overground gait (phase) analysis dorsal steps were identified only for the left hindlimb and any dorsal steps on the right side would not be identified in the analyzed data set. In separate analysis (CPI, DSI, right-left dorsal step ratio) dorsal steps for both right and left hindpaws were considered (see Figure 7).”

This is shown highlighted in the manuscript.

Please review all figure legends to be sure that the meaning of the asterisks shown in the figures is clearly defined.

Asterisks are now clearly defined. Thanks for pointing this out.

Reviewer #2 (Recommendations for the authors):It was not evident why there was not more of a focus on homolateral and heterolateral forelimb/hindlimb coupling. When one thinks about LDPNs, the obvious changes would be expected to be in the forelimb-hindlimb coupling, rather than left-right at lumbar and cervical levels. A disruption in either homolateral or heterolateral coupling is also likely to disrupt left-right coupling. It is understandable that left/right coupling is featured prominently due to the results from the LAPNs but forelimb-hindlimb coupling should also be displayed. Based on Figure 2 supp 1, it looks like heterolateral coupling is disrupted by silencing but not homolateral.

We agree that the expectations of FL-HL coordination being disrupted is logical. However, our data overall supports the concept that R-L phase at each girdle is controlled at least partially independent of FL-HL coordination. Thus, if R-L phase is disrupted then any/most of the FL-HL coordination changes would be secondary consequences, rather than primary. If FL-HL coordination were the primary disruption, we would anticipate that R-L phase at each girdle would be intact. In response to this concern we have added a figure 2 figure supplement 2 that shows heterolateral and homolateral coupling. In the text we will revisit the idea that FL-HL disruption appears to be secondary to the disruption at each girdle.

“During silencing, the disruption to interlimb coordination included modest but significant changes in heterolateral and homolateral forelimb-hindlimb coupling, as observed previously when silencing the LAPNs (Figure 2 figure supplement 1; Pocratsky et al., 2020). Based on the magnitude of the effect and the apparent hierarchy of impact, we consider the disruption of forelimb-hindlimb coupling to be secondary to the partial de-coupling of the forelimb and hindlimb pairs (see discussion).”

And later

“The spaces in these graphs, most notably centrally in D and J, and middle top and bottom in E and K, also support the concept that forelimb-hindlimb coordination was only disrupted when the forelimb and hindlimb pairs were partially de-coupled and thus was a secondary rather than primary effect of silencing.”

Added to the Results section. Also,

“Further, disrupted forelimb-hindlimb coordination was observed only when the right-left alternation of the forelimb and hindlimbs occurred, suggesting that the latter is the primary impact of silencing while the former a secondary effect.”

Was added to the discussion. Both additions are shown highlighted in the manuscript.

The single-figure examples of terminals in Figure 6 are not entirely convincing. Some sort of quantification of terminals should be possible in both SCI and uninjured cords. It is also difficult to interpret that data without knowing the approximate percentage of LDPNs that were silenced (or that expressed TeTN).

We agree that these images are not very satisfying. We would truly love to be able to provide the requested quanitfication, but it is not feasible. The eTeTN.GFP expression/staining does not seem to be as reliable as the silencing itself and this is the one major drawback of this system (in our opinions). We would make the argument that the phenotype is comparatively robust given the relatively small number of LAPNs and LDPNs (see Brown et al., 2022 PMID: 33828464 for LAPN anatomy) and that demonstrating eTeTN.GFP+ fibers in the target region (C6 intermediate gray matter) is sufficient to be confident that the phenotype results from the synaptic silencing of LDPNs rather than some other mechanism. We acknowledge that the LAPNs may also have synapses outside of C6 and in other work we are tackling a comprehensive anatomical analysis in both intact and injured spinal cord (Brown et al. is the first step in that process). We hope this is sufficient even if not ideal.

Reviewer #3 (Recommendations for the authors):This study explores the role of descending propriospinal neurons on inter-limb coordination during locomotion and how their silencing helps improve some parameters after spinal cord injury. Below are specific concerns and comments on this study.Results1) Line 96: please clarify how the double infection was performed, which viruses were injected and how did you ensure that neurons were silenced. Figure 1 only outlines the experimental procedure and does not provide any data in this sense. How do the authors control that TeNT is reversible and how long does it take to reverse the silencing?

See comments above re: reviewer 1 (#1).

2) Were the injected viruses tagged with fluorescent reporters? It would be helpful to provide some quantification on the number of DLPNs neurons and their distribution in the spinal cord. Are they restricted to C6?

See comments above re: reviewer 2 (#2)

3) Figure 2: please define the abbreviations used in the graphs. Why were values from each trial in the same animal presented rather than the mean value per animal? Is this statistically correct?

Data/results were presented in this way in order to be fully transparent with the concept that if the animal expressed a certain pattern, even once, it is relevant. We use rigorous and appropriate statistical analysis throughout as handled by our full time statistician, Ms Darlene Burke.

4) It is not clear to this reviewer why and how silencing of DLPNs in lumbar segments affects the coordination of forelimbs.

This is one of the most interesting aspects of this work, and we provide significant discussion on this matter. We believe that the LA and LDPNs form an ascending-descending loop that can influence the interlimb coordination at each girdle and as we have shown, this occurs when either the LA or LDPNs are silenced.

4) Figure 3, graphs A-J: it is more relevant to plot data points from control Acryl and Dox Acryl together and control Sylgard and Dox Sylgard together in the same graphs. This will allow the reader the appreciate potential differences. Here again, it is not clear why data points from each trial were presented rather than mean values per animal.

We have chosen to present the acryl vs sylgard for Control and Dox conditions in order to emphasize any context specificity that exists (does walking surface influence the phenotype?), as this is most relevant to interpretation of the data, from our perspective. By placing the control and dox graphs above/below each other we believe a visual comparison can be made. Part of the reason we present individual data points also has to do with convention and how locomotor control data/results are traditionally displayed. Finally, the averages for these kinds of data/results tend to hide aspects of the pattern of change (with speed, for example) which are easy to appreciate when presented as we have in Figure 3. We hope this explanation is sufficient. We would, if asked, be happy to present these data as averages in a table, if that would help.

5) Figure 5: the figure and text need more clarification. The authors should indicate in the different panels the parts of the white matter that have been spared and the location of the descending projections. It is not clear how the authors come to the conclusion that the LDPN axons have been spared while only 26% of white matter was spared.

We apologize for the too-definitive statements. We did not intend to support the idea that the LDPNs are spared en mass. We have edited a number of statements throughout the paper to indicate that some proportion are spared because they reside in the most lateral parts of the lateral and ventrolateral white matter. We have also added asterix to two of the images in figure 5 to illustrate the locations of spared lateral and ventrolateral white matter.

6) Figure 6: The authors state that GFP is expressed in cell bodies of neurons in the cervical spinal cord. However, the staining looks more like terminals contacting the stained cell bodies rather than the expression of GFP. Quantifications are necessary here to support the authors' conclusions.

We apologize for the confusing wording. The legend for figure 6 has been edited as requested. In G and H, the staining is thought to be intracellular compartments positive for eGFP. As explained earlier, quantification of the eTeNT.GFP is not feasible due to inconsistency. This has been described previously by Isa and colleagues who developed the two-virus silencing system and is why we are pursuing separate virus-based tracing strategies (e.g. Brown et al., 2022). We hope this is acceptable.

Discussion4) In the discussion, the authors state that the study by Ruder et al. (2016) used developmental ablation of subsets of LDPNs. Ruder et al. used viral infections to express DTR in descending propriospinal neurons in adult mice. The difference in the outcome of this study compared to that of Ruder et al. cannot be explained by circuit reorganization during development. In the present study, there is no information on the extent of silencing and which neuronal populations are affected. This part of the discussion does not reflect the existing data and does not explain the difference in the outcome between this study and Ruder et al. (2016).

Thank you for catching this serious error. The study by Ruder et al. does have the limitation of using ablation, which is not reversible and which may trigger a plastic response to the cell death. Nonetheless, we misrepresented the paper and this has been corrected. We agree that we do not know the extent of silencing, but I contend we do know which neuronal populations are silenced, they are those defined anatomically by cell bodies at C6 and terminals at L2. The paragraph in question has been carefully edited.

5) Overall, it is not clear how silencing of DLPNs restores some parameters after spinal cord injury and how relevant this approach would be clinically.

We agree, and did not intend that silencing could in any way be used clinically. However, we do believe that our surprising results are relevant for therapeutic development, in particular with respect to neuromodulation (epidural stimulation) that may actively target circuitry that bridges the area of an incomplete injury.

[Editors’ note: what follows is the authors’ response to the second round of review.]

The manuscript has been improved but there are some remaining issues that need to be addressed, as outlined below:1. Please carefully check and revise the text discussing how your findings relate and compare to those reported by Ruder (2016). In particular, please note that in the Ruder paper, some experiments looking at neural circuits were conducted based upon tamoxifen-induced Cre-mediated recombination at E10.5 (Figure 1) but it appears to us that experiments examining functional outcomes were after manipulating gene expression by injecting viral constructs after birth. Thus, when discussing how ablating subsets of LDPNs please be sure you are referencing and discussing the correct experimental system.

Thank you for bringing this to our attention. Clearly, we confused the Ruder et al., 2016 paper with other work that focused on developmental ablations. We have edited the paragraph in question to much more clearly discuss the similarities and differences between our work and the work from the Arber lab. We also made a minor edit to the following paragraph to include a reference to the Ruder et al., 2016 paper which adds strength to the discussion. The two paragraphs now read as follows:

“Ruder et al. (2016) reported that in the adult mouse, ablation of LDPNs using intersectional approaches reduced the speed and duration of spontaneous locomotor bouts, and disrupted interlimb coordination during high-speed locomotion on a treadmill. In contrast, our data indicate that, despite the disrupted alternation of the two limb pairs, rats are still able to achieve fairly high rates of speed and that locomotor bouts are sustained and unperturbed when LDPNs are silenced in otherwise intact adults. These discrepancies may arise from several differences between the two studies. First, the specific neurons targeted by Ruder et al. may have included LDPNs from multiple cervical segments whereas we chose to focus specifically on LDPNs with cell bodies at C6. Second, ablation of neurons using the genetic approach (with diptheria toxin) is permanent and may lead to unrecognized circuit reorganization in the adult.

The LDPN population targeted in the current study may be comprised of different sub-populations of long spinal projection neurons that have been previously defined genetically in mice. Notably, V0v and V2a, but not V0d, neurons residing in the lumbar enlargement have been implicated in securing left-right limb alternation at high speeds (Bellardita and Kiehn 2015; Crone et al. 2009; Talpalar et al. 2013; Ruder et al., 2016). V0v are composed of commissurally-projecting inhibitory interneurons and are recruited to maintain left-right alternation as the speed of movement increases (Lanuza et al. 2004; Crone et al. 2009). The V2a population of neurons are also classified as ipsilateral excitatory glutamatergic interneurons that are involved in left-right alternation (Crone et al. 2008; 2009) with minimal involvement in rhythm generation. It is likely that the silenced LDPN population includes some proportion of each of these defined interneurons, given the perturbations to left-right alternation during silencing. However, it is important to note that LDPNs have characteristics that fall in line with both progenitor domains, as they are both ipsilaterally and commissurally-projecting and co-localize with both glutamatergic and GABAergic synapses. Therefore, it is difficult to place these anatomically-defined neurons within the previously described context of genetically-defined V-neuron classes.”